# Neuronal Learning Analysis using Cycle-Consistent Adversarial Networks

## Abstract

Understanding how activity in neural circuits reshapes following task learning could reveal fundamental mechanisms of learning. Thanks to the recent advances in neural imaging technologies, high-quality recordings can be obtained from hundreds of neurons over multiple days or even weeks. However, the complexity and dimensionality of population responses pose significant challenges for analysis. Existing methods of studying neuronal adaptation and learning often impose strong assumptions on the data or model, resulting in biased descriptions that do not generalize. In this work, we use a variant of deep generative models called – cycle-consistent adversarial networks, to learn the unknown mapping between pre- and post-learning neuronal activities recorded *in vivo*. To do so, we develop an end-to-end pipeline to preprocess, train and evaluate calcium fluorescence signals, and a procedure to interpret the resulting deep learning models. To assess the validity of our method, we first test our framework on a synthetic dataset with known ground-truth transformation. Subsequently, we applied our method to neuronal activities recorded from the primary visual cortex of behaving mice, where the mice transition from novice to expert-level performance in a visual-based virtual reality experiment. We evaluate model performance on generated calcium imaging signals and their inferred spike trains. To maximize performance, we derive a novel approach to pre-sort neurons such that convolutional-based networks can take advantage of the spatial information that exists in neuronal activities. In addition, we incorporate visual explanation methods to improve the interpretability of our work and gain insights into the learning process as manifested in the cellular activities. Together, our results demonstrate that analyzing neuronal learning processes with data-driven deep unsupervised methods holds the potential to unravel changes in an unbiased way.

## 1 Introduction

One of the main objectives in computational neuroscience is to study the dynamics of neural processing and how neural activity reshapes in the course of learning. A major hurdle was the difficulty in obtaining high-quality neural recordings of the same set of neurons across multiple experiments, though such limitation in recording techniques has seen tremendous improvements in recent years. With the advent of modern neural imaging technologies, it is now possible to monitor a large population of neurons over days or even weeks (Williams et al., 2018a; Steinmetz et al., 2021), thus allowing experimentalists to obtain *in vivo* recordings from the same set of neurons across different learning stages. Significant efforts have been put into extracting interpretable and unbiased descriptions of how cortical responses change with experience. Proposed approaches to model changes in neuronal activity include linear latent variable models such as PCA, TCA, GPFA, GPFADS and PSID (Cunningham & Byron, 2014; Williams et al., 2018b; Sani et al., 2021; Yu et al., 2009; Rutten et al., 2020). Methods employing deep learning models but with linear changes or mapping include LFADS and PfLDS (Pandarinath et al., 2018; Gao et al., 2016). While these methods enabled substantial progress in understanding the structure of neuronal activity, they do have strong assumptions inherent in the modelling technique or the analysis, such as the linearity assumption in linear latent variable models. Therefore, making sense of the unknown mapping between pre- and post-learning neural activity in an unbiased manner remains a significant challenge, and a data-driven method to interpret the circuit dynamics in learning is highly desirable.

Thanks to their ability to self-identity and self-learn features from complex data, deep neural networks (DNNs) have seen tremendous success in many biomedical applications (Cao et al., 2018; Zemouri et al., 2019; Piccialli et al., 2021). Specifically, deep generative networks have shown promising results in analyzing and synthesizing neuronal activities in recent years. Pandarinath et al. (2018) developed a variational autoencoder (VAE) to learn latent dynamics from single-trial spiking activities and Prince et al. (2020) extended the framework to work with calcium imaging data. Numerous work have demonstrated generative adversarial networks (GAN) are capable of synthesizing neuronal activities that capture the low-level statistics of recordings obtained from behaving animals (Molano-Mazon et al., 2018; Ramesh et al., 2019; Li et al., 2020).

In this work, we explore the use of cycle-consistent adversarial networks (Zhu et al., 2017), or CycleGAN, to learn the mapping between pre- and post-learning neuronal activities in an unsupervised and data-driven manner. In other words, given the neural recordings of a novice animal, can we translate the neuronal activities that correspond to the animal with expert-level performance, and vice versa? The resulting transformation summarizes these changes in response characteristics in a compact form and is obtained in a fully data-driven way. Such a transformation can be useful in follow-up studies to 1) identify neurons that are particularly important for describing the changes in the overall response statistics, not limited to first or second order statistics; 2) detect response patterns relevant for changes from pre- to post-learning; and 3) determine what experimental details are of particular interest for learning.

To learn the transformation, we derive a standardized procedure to train, evaluate and interpret the CycleGAN framework. To improve the explainability of our work, we incorporate a self-attention mechanism into our generator models and also employ a feature-importance visualization method into our pipeline so that we can visualize and identify the input that the networks deemed relevant in their decision making process. In addition, we introduced a novel neuron ordering method to improve the learning performance of convolutional neural networks (CNN). To quantify the capability of the proposed unsupervised learning method, we evaluate our method on two datasets: 1) an artificially constructed dataset with a handcrafted transformation, and 2) recordings obtained from the primary visual cortex of a behaving animal across multiple days. We then compare several metrics and statistics between the recorded and translated calcium traces and their inferred spike trains.

## 2 METHODS

2.1 ANIMAL EXPERIMENT   To obtain neuronal activities that can demonstrate pre- and post-learning responses, we conducted a visual-based experiment which follows a similar procedure as Pakan et al. (2018) and Henschke et al. (2020). Briefly, a head-fixed mouse was placed on a linear treadmill that allows it to move forward and backward. A lick spout and two monitors were placed in front of the treadmill and a virtual corridor with defined grating pattern was shown to the mouse. A reward (water drop) would be made available if the mouse licked within the predefined reward location (at 120-140 cm), in which a black screen is displayed as a visual clue. Figure A.1 illustrates the experiment setup. The mouse should learn to utilize both visual information and self-motion feedback to maximize reward. The same set of neurons in the primary visual cortex were labelled with GCaMP6 calcium indicator and monitored throughout 4 days of experiment, the relative changes in fluorescence ($\Delta F/F_0$) over time were used as a proxy for an action potential. 4 mice were used in the virtual-corridor experiment and all mice transitioned from novice to expert in the behaviour task within 4 days of training. Mouse 1 took on average 6.94s per trial on day 1 and 4.43s per trial on day 4, Table A.1 and A.2 shows the trial information of all the mice. Hence, this dataset can provide excellent insights into how cortical responses reshape with experience.

2.2 CYCLEGAN   CycleGAN (Zhu et al., 2017) is a GAN-based unsupervised framework that learns the mapping between two unpaired distributions $X$ and $Y$ via the adversarial training and cycle-consistency optimization. The framework has shown excellent results in a number of unsupervised translation tasks, including natural language translation (Gomez et al., 2018) and molecular optimization (Maziarka et al., 2020), to name a few.

Let $X$ and $Y$ be two distributions with unknown mappings that correspond to (novice) pre- and (expert) post-learning neuronal activity, respectively. CycleGAN consists of four DNNs: generator $G : X \rightarrow Y$ that maps novice activities to expert activities and generator $F : Y \rightarrow X$ that maps expert activities to novice activities; discriminator $D_X : X \rightarrow [0, 1]$ and discriminator $D_Y : Y \rightarrow [0, 1]$ that learn to distinguish novice and expert neural activities, respectively. In a forward

cycle step ($X \rightarrow Y \rightarrow X$, illustrated in Figure B.1), we first sample a novice recording $x$ from distribution $X$ and apply transformation $G$ to obtain $\hat{y} = G(x)$. We expect $\hat{y}$ to resembles data from the expert distribution $Y$, hence $D_Y$ learns to minimize (1) $\mathcal{L}^{D_Y} = -\mathbb{E}_{y \sim Y}[(D_Y(y) - 1)^2] + \mathbb{E}_{x \sim X}[D_Y(G(x))^2]$. Similar to a typical GAN, generator $G$ learns to deceive $D_Y$ with the objective of (2) $\mathcal{L}^G = -\mathbb{E}_{x \sim X}[(D_Y(G(x)) - 1)^2]$. Note that these are same objectives in LSGAN (Mao et al., 2017). However, $D_Y$ can only verify if $\hat{y} \in Y$, though cannot ensure that $\hat{y}$ is the corresponding expert activity of the novice recording $x$. Moreover, $X$ and $Y$ are not paired hence we cannot directly compare $\hat{y}$ with samples in $Y$. To tackle this issue, CycleGAN applies another transformation to reconstruct the novice recording $\bar{x} = F(\hat{y})$ where the distance $\| x - \bar{x} \|$ or $\| x - F(G(x)) \|$ should be minimal. Therefore, the generators also optimize this cycle-consistent loss (3) $\mathcal{L}_{\text{cycle}} = \mathbb{E}_{x \sim X}[\| x - F(G(x)) \|] + \mathbb{E}_{y \sim Y}[\| y - G(F(y)) \|]$. Mean absolute error (MAE) was used as the distance function, though other distance functions can also be employed. In addition, we would expect $\hat{x} = F(x)$ and $\hat{y} = G(y)$ to be in distributions $X$ and $Y$ given that $F : Y \rightarrow X$ and $G : X \rightarrow Y$, hence the identity loss objective (4) $\mathcal{L}_{\text{identity}}^G = \mathbb{E}_{y \sim Y}[\| y - G(y) \|]$.

Taken all together, $G$ optimizes the following objectives: (5) $\mathcal{L}_{\text{total}}^G = \mathcal{L}^G + \lambda_{\text{cycle}} \mathcal{L}_{\text{cycle}} + \lambda_{\text{identity}} \mathcal{L}_{\text{identity}}^G$ where $\lambda_{\text{identity}}$ and $\lambda_{\text{cycle}}$ are hyper-parameters for identity and cycle loss coefficients. All four networks are trained jointly where $\mathcal{L}_{\text{total}}^F$ and $\mathcal{L}^{D_X}$ are similar to $\mathcal{L}_{\text{total}}^G$ and $\mathcal{L}^{D_Y}$ though in opposite directions. In this work, we adapt the CycleGAN framework to learn the unknown mapping between pre- and post-learning neuronal activities recorded from the primary visual cortex of behaving mice. In addition, we experiment with different GANs objective formulations on top of the original LSGAN objective, including GAN (Goodfellow et al., 2014), WGANGP (Arjovsky et al., 2017) and DRAGAN (Kodali et al., 2017). Table C.2 shows their exact formulations in CycleGAN.

2.3 MODEL PIPELINE    We devise a consistent analysis framework, including data preprocessing and augmentation, networks interpretation, and evaluation of the generated calcium fluorescence signals and their inferred spike trains. Figure C.1 illustrates the complete pipeline of our work.[1]

We denote the day 1 (pre-learning) and day 4 (post-learning) recording distributions to be $X$ and $Y$. With Mouse 1, $W = 102$ neurons from the primary visual cortex were monitored, as well as trial information such as the virtual distance, licks and rewards. In total, 21471 and 21556 samples were recorded on day 1 and 4. Since we want the generators and discriminators to identify patterns relevant to the animal experiment in a data-driven manner, we do not incorporate any trial information into the training data. We first segment the two datasets with a sliding window of size $H = 2048$ along the temporal dimension (around 85 s in wall-time), resulting in data with shape $(N, H, W)$ for $X$ and $Y$ where $N$ is the total number of segments. We select a stride size that space out each segment evenly so that we obtained a sufficient number of samples while keeping the correlations between samples reasonably low. In order to take advantage of the spatiotemporal information in the neuronal activities in a 2D CNN, we further convert the two sets to have shape $(N, H, W, C)$ where $C = 1$. Finally, we normalize each set to the range $[0, 1]$, and divide them into train, validation and test set with 3000, 200 and 200 samples respectively.

To evaluate the transformation results of $G$ and $F$, we can compare the cycle-consistency MAE$(X, F(G(X)))$ and MAE$(Y, G(F(Y)))$, as well as the identity losses MAE$(X, F(X))$ and MAE$(Y, G(Y))$ (e.g. we expect $F$ to apply no transformation to a novice sample $x$). We also evaluate the generated data in terms of spike activities in the following distribution combinations: novice against translated novice ($X \mid F(Y)$), novice against reconstructed novice ($X \mid F(G(X))$), expert against translated expert ($Y \mid G(X)$) and expert against reconstructed expert ($Y \mid G(F(Y))$). We use Cascade (Rupprecht et al., 2021) to infer spike trains from the recorded and generated calcium signals to assess the credibility of the generated signals. We measure the following commonly used spike train similarities and statistics: 1) mean firing rate for evaluating single neuron statistics; 2) pairwise Pearson correlation for evaluating pairwise statistics; 3) pairwise van Rossum distance (Rossum, 2001) for evaluating general spike train similarity. We evaluate these quantities across the whole population for each neuron or neuron pairs and compare the resulting distributions over these quantities obtained from the recorded and generated data. We, therefore, validate the whole spatiotemporal first and second-order statistics as well as general spike train similarities.

To improve the explainability of this work we introduce a number of recently proposed model interpretation methods into our pipeline. We design a self-attention generator architecture which allows

---

[1]The software codebase will be made publicly available upon acceptance.

the network to learn a set of attention masks such that it encourages the network to better focus on specific areas of interest in the input and also enables us to visually inspect the learned attention maps. In addition, we use GradCAM (Selvaraju et al., 2017), a method to visualize discriminative region(s) learned by a CNN classifier w.r.t to the input, to extract localization maps from the generators and discriminators. The self-attention mechanism and GradCAM visualization allow us to verify and interpret that the networks are learning meaningful features. Moreover, these extracted attention maps can reveal neurons or activity patterns that are informative in the neuronal learning process. A detail description of the model architectures are available in Section D.

2.3.1 NEURON ORDERING    CNNs with a smaller kernel can often perform as well or even better than models with larger kernels while consisting of fewer trainable parameters (He et al., 2016a; Li et al., 2021). Nevertheless, a smaller kernel can also limit the receptive field of the model, or the region in the input that the model is exposed to in each convolution step (Araujo et al., 2019). In addition, the recordings obtained from the virtual-corridor experiment were annotated based on how visible the neurons were in the calcium image, rather than ordered in a particular manner (see Figure A.1). This could potentially restrict CNNs with small receptive field to learn meaningful spatial-temporal information from the population responses. To mitigate this issue, we derive a procedure to pre-sort $X$ and $Y$, such that neurons that are highly correlated or relevant are nearby in their ordering. A naive approach is to sort the neurons by their firing rate or average pairwise correlation, where the neuron with the highest firing rate or the neuron that, on average, is most correlated to other neurons is ranked first in the data matrix. However, it is possible that not all high-firing neurons or most correlated neurons are the most influential in the learning process. Therefore, we also explore a data-driven approach. Deep autoencoders have shown excellent results in feature extraction and representation learning (Gondara, 2016; Wang et al., 2016; Tschannen et al., 2018), and we can take advantage of its unsupervised feature learning ability.

We employ a deep autoencoder `AE` which learns to reconstruct calcium signals in $X$ and $Y$ jointly. `AE` consists of 3 convolution down-sampling blocks, followed by a bottleneck layer, then 3 transposed-convolution up-sampling blocks. The down-sampling block consists of a convolution layer followed by Instance Normalization (Ulyanov et al., 2016), Leaky ReLU (LReLU) activation (Maas et al., 2013) and Spatial Dropout (Tompson et al., 2015), whereas a transpose convolution is used in the up-sampling block instead. We optimize the mean-squared error (`MSE`) reconstruction loss on the training set of $X$ and $Y$, then we use the per-neuron reconstruction error on the test set to sort the neurons (in ascending order): order $= \text{argsort}(0.5 \times [\,\text{MSE}(X, \text{AE}(X)) + \text{MSE}(Y, \text{AE}(Y))\,])$. The neuron sorting process is part of the data preprocessing step and is independent from the Cycle-GAN framework.

2.3.2 SYNTHETIC DATA    CycleGAN was originally introduced for image-to-image translation. Albeit the two image distributions are not aligned hence cannot be directly compared easily, one could still visually inspect whether or not $\hat{x} = F(y)$ and $\hat{y} = G(x)$ are reasonable transformations. However, it would be difficult to visually inspect the two transformations with calcium signals. To this end, we introduce an additional dataset $Y = \Phi(X)$ with a known transformation $\Phi$, such that $G : X \rightarrow Y = \Phi(X)$ and $F : Y = \Phi(X) \rightarrow X$. We can then verify $G(x) = \hat{y} = \Phi(x)$ and $F(y) = \hat{x} = x$. We defined the spatiotemporal transformation $\Phi$ that can be identified visually as follows: (6) $\Phi(x) = m_{\text{diagonal}}x + 0.5\eta$, where $m_{\text{diagonal}}$ is a diagonal mask to zero-out the lower left corners of the signals and $\eta \sim \mathcal{N}(\mu_x, \sigma_x^2)$. $\mu_x$ and $\sigma_x$ are the per-neuron mean and standard deviation of $X$. Figure 1 shows an augmented example. Importantly, we shuffle the train set after the augmentation procedure so that $X$ and $Y$ appears to be unpaired to the model. Whereas the test set remains in its original paired arrangement so that we can compare $\parallel X - F(Y) \parallel$ and $\parallel Y - G(X) \parallel$.

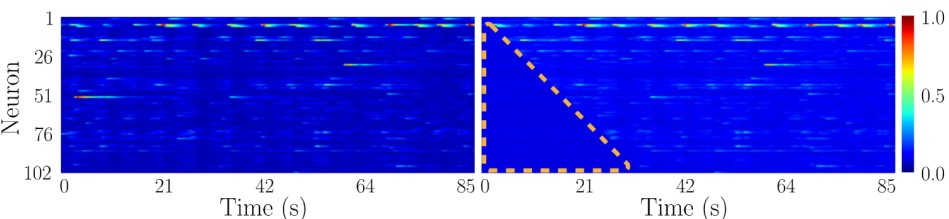

Figure 1: (Left) original $x$ and (Right) augmented $y = \Phi(x)$ calcium traces, where the bottom left corner (yellow dashed triangle) in $y$ has been masked out and noise being added to the segment.

## 3 RESULTS

We assessed the CycleGAN framework on synthetic data with known ground truth and on experimental data where we recovered trial information. We also experimented with different GAN objective formulations as well as different neuron ordering methods. All models presented below were trained with the Adam optimizer (Kingma & Ba, 2014) for 200 epochs where all models converged. We trained all CycleGAN models on a single NVIDIA A100 GPU which on average took 15 hours to complete. It took an additional hour to train the autoencoder in the case where we pre-sort neurons according to the `AE` reconstruction loss. Table C.1 details the hyper-parameters used.

3.1 SYNTHETIC DATA   To show that our method is capable of learning subtle differences in calcium traces, we first fit our model on the synthetic dataset. Figure 2 shows calcium signals of the forward and backward cycle transformation of neuron 75 from a randomly selected test segment, where `AGResNet` generators were trained with LSGAN objectives (more examples in Figure F.1). Without paired samples, $F(y)$ made a reasonable attempt in reconstructing the augmented region in $y$, whereas $G(x)$ was able to learn to mask out the appropriate regions in $x$. Since $\Phi(X) = Y$ performed a systematic spatiotemporal transformation to $X$, one would expect the networks to learn features that focus on the augmented region of the data. We, therefore, use GradCAM (Selvaraju et al., 2017) localization maps to visualize regions of interest learned by the discriminators. The localization map of discriminator $D_Y(Y)$ when given an augmented sample $y = \Phi(x)$, shown in Figure 3, demonstrates a high level of attention around the edge of the diagonal region. This indicates that $D_Y$ learned to distinguish whether or not a given sample is from distribution $Y = \Phi(X)$ by predominantly focusing on the edge of the masking area. On the other hand, since no augmentation was done on the input to discriminator $D_X$, the localization map does not appear to have a particular structural area of focus at first (c.f. Figure F.3). Interestingly, once we overlay the reward zones on the input, we observe that the area of focus learned by $D_X$ is loosely aligned with the reward zones. Note that reward zones are external task-relevant regions that are expected to shape the neural activity in the primary visual cortex as the visual patterns change when the mouse enters the reward zone. Our findings therefore suggest that $D_X$ learned distinctive patterns from highly ranked neurons around the reward zones. Figure 3 shows the AG sigmoid masks from $G(x)$. Both attention masks ignored the augmentation region (i.e. bottom left corner), as information in that area is not relevant in the $G : X \rightarrow \Phi(X)$ transformation. Similar, $F$ which should learn $\Phi(X) \rightarrow X$ also allocated less focus in the masked region in its reconstruction process, as it contains no useful information. (see Figure F.3).

Since $X$ and $Y = \Phi(X)$ are paired in the test set, this allows us to compute $\text{MAE}(Y, G(X))$ and $\text{MAE}(X, F(Y))$ hence providing a good testbed to compare different generator architectures, GAN objective formulations and neuron ordering methods. We also added the identity models as baseline, which should have perfect cycle-consistent loss as $F(G(x))$ and $G(F(y))$ perform no operation on the data. Nevertheless, despite the fact that $\Phi$ is a relatively simple augmentation, one would expect the difference between $X$ and $\Phi(X)$ to be small. Table 1 shows the direct comparison results of different combinations of objective formulations, generator architectures and neuron ordering methods. Both `ResNet` and `AGResNet` achieved significantly better results than the identity model. To mitigate the issues of vanishing gradient and mode collapse, we used gradient penalty regularization to enforce the 1-Lipschitz condition in the discriminator. We, therefore, tested 4 popular GAN objectives with the CycleGAN framework. Interestingly, the LSGAN objectives achieved slightly better results than GAN objectives while both performed better than identity. The two objectives with gradient penalty obtained lower cycle-consistent errors than GAN and LSGAN, yet performed significantly worse in the intermediate transformations $F(Y)$ and $G(X)$. This suggests that the discriminators could be overpowered by the generators when trained with WGANGP and DRAGAN, in which $D_X(F(Y))$ and $D_Y(G(X))$ are neither informative nor impactful to the overall objective. This is likely because the gradient penalty regularization further complicates the already perplexing CycleGAN objectives. We employed 3 different methods to pre-sort neurons in the data, including firing rate, pairwise correlation and autoencoder reconstruction loss. In addition, to demonstrate that 2D convolution can indeed better learn the spatial structure in neuronal activities, we trained a 1D variant of `AGResNet` (denoted as `1D-AGResNet`) as baseline which disregards all spatial information. Overall, models trained on sorted neurons achieved better results compared to unordered neurons and in most cases, sorting neurons according to the autoencoder reconstruction loss performed the best. Moreover, `1D-AGResNet` performed significantly worse than its 2D counterparts, suggesting that the spatial structure in the neural activities is indeed important. In the

remaining work, we use the LSGAN objective to train the generators with the `AGResNet` architecture along with neurons ordered based on autoencoder reconstruction loss as this combination achieved the best overall results on the synthetic data.

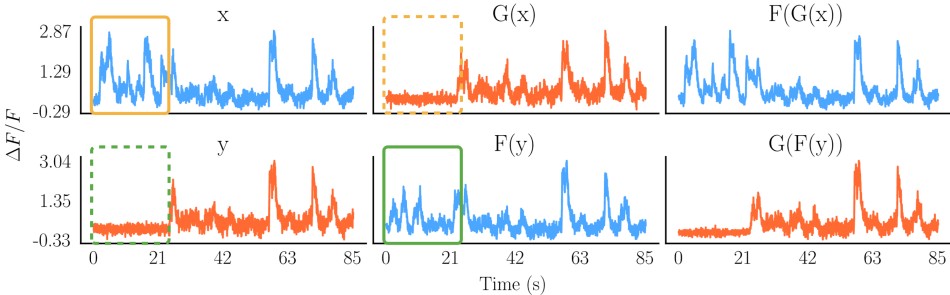

Figure 2: Forward and backward cycle steps of neuron 75 from a randomly selected test segment. $G$ should learn to translate signals in the yellow solid box to yellow dotted box, and $F$ from green dotted box to green solid box. We expect the signals in the green solid box resemble signals in the yellow solid box and yellow dotted box to green dotted box. More examples are shown in Figure F.1

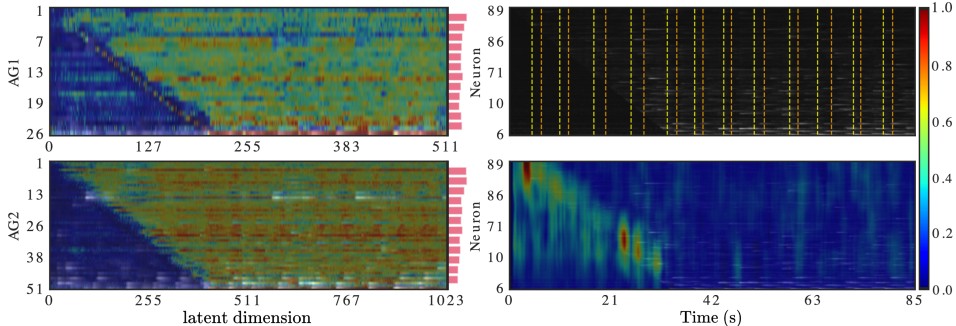

Figure 3: (Left) Learned attention masks $AG_1$ and $AG_2$ in `AGResNet` $G$ given a random test segment $x \sim X$. $AG_1$ and $AG_2$ both learned to ignore information in (to-be) masked region in $x$. Note that $AG_1$ and $AG_2$ are 4 and 2 times lower-dimensional than the original input dimension. (Right) GradCAM localization map of $D_Y$ given a randomly select test segment $y = \Phi(x) \sim X$. The top panel shows the original input, where yellow and orange dotted lines mark the start and end of each reward zone. The second panel shows the GradCAM localization map superimposed on the input. Neurons were ordered based on `AE` reconstruction loss, the exact ordering is available in Table E.1. Figure F.3 shows the attention gates and GradCAM map of $F(y)$ and $D_X(x)$.

3.2 RECORDED DATA    As our proposed method has successfully learned the unpaired transformations in the synthetic dataset, we now move on to the recordings obtained from the virtual-corridor experiment where we attempt to learn the unknown mapping between pre- and post-learning neuronal activity. Figure 4 shows the cycle transformation of neuron 50 from a randomly selected test segment. Visually, $G$ and $F$ seems to be able to reconstruct $\bar{x} = F(G(x))$ and $\bar{y} = G(F(y))$, and that the two generators are not simply passing through $x$ and $y$ in intermediate step $\hat{y} = G(x)$ and $\hat{x} = F(y)$. To better analyse the transformation performance, we first compare the generated calcium florescence signals with the recorded test set data. The cycle-consistent loss on the test set achieved a values of $\texttt{MAE}(X, F(G(X))) = 0.0733$ and $\texttt{MAE}(Y, G(F(Y))) = 0.0737$. The identity losses for $\texttt{MAE}(Y, G(Y))$ and $\texttt{MAE}(Y, G(Y))$ are also minimal, with values of $0.0101$ and $0.0069$, respectively. For reference, $\texttt{MAE}(X, Y) = 0.3674$. This suggest $G$ and $F$ are not simply passing through the data without any processing. In addition, the low identity loss indicates that the generators can correctly identify whether or not the given input is already part of its target distribution. Table G.1 reports the cycle-consistent and identity loss with different neuron ordering methods.

Since we lack paired data in the *in vivo* recordings, we cannot directly compare $\texttt{MAE}(X, F(Y))$ nor $\texttt{MAE}(Y, F(X))$, in contrast to Section 3.1. In order to better analyse the two intermediate transformations $\hat{y} = G(x)$ and $\hat{x} = F(y)$, and show that $G$ and $F$ can indeed translate $x$ and $y$ into their respective distributions $\hat{y} \sim Y$ and $\hat{x} \sim X$, we also compare a set of spike train statistics. Section H

| | $|X - F(Y)|$ | $|X - F(G(X))|$ | $|Y - G(X)|$ | $|Y - G(F(Y))|$ |
|---|---|---|---|---|
| (A) DIFFERENT MODELS WITH LSGAN OBJECTIVE | | | | |
| IDENTITY | $0.4234 \pm 0.0172$ | **0** | $0.4234 \pm 0.0172$ | **0** |
| RESNET | $0.1617 \pm 0.0071$ | $0.1173 \pm 0.0043$ | $0.3743 \pm 0.0391$ | $0.1247 \pm 0.0067$ |
| AGRESNET | $\mathbf{0.1508 \pm 0.0089}$ | $0.1107 \pm 0.0051$ | $\mathbf{0.2520 \pm 0.0262}$ | $0.1467 \pm 0.0084$ |
| (B) DIFFERENT OBJECTIVES WITH AGRESNET | | | | |
| GAN | $0.1611 \pm 0.0063$ | $0.0948 \pm 0.0069$ | $0.2513 \pm 0.0350$ | $0.1491 \pm 0.0050$ |
| LSGAN | $\mathbf{0.1508 \pm 0.0089}$ | $0.1107 \pm 0.0051$ | $\mathbf{0.2520 \pm 0.0262}$ | $0.1467 \pm 0.0084$ |
| WGANGP | $0.2381 \pm 0.0123$ | $0.1600 \pm 0.0098$ | $0.3186 \pm 0.0096$ | $0.1960 \pm 0.0093$ |
| DRAGAN | $0.3832 \pm 0.0115$ | $\mathbf{0.0434 \pm 0.0021}$ | $0.4012 \pm 0.0207$ | $\mathbf{0.0568 \pm 0.0027}$ |
| (C) DIFFERENT NEURON ORDERING WITH AGRESNET AND LSGAN OBJECTIVE | | | | |
| 1D-AGRESNET | $0.2724 \pm 0.0101$ | $0.1878 \pm 0.0115$ | $0.3151 \pm 0.0445$ | $0.1655 \pm 0.0115$ |
| ORIGINAL | $0.1508 \pm 0.0089$ | $0.1107 \pm 0.0051$ | $0.2520 \pm 0.0262$ | $0.1467 \pm 0.0084$ |
| FIRING RATE | $0.1578 \pm 0.0079$ | $0.0722 \pm 0.0044$ | $0.1304 \pm 0.0306$ | $0.0842 \pm 0.0036$ |
| CORRELATION | $0.1556 \pm 0.0044$ | $0.0852 \pm 0.0042$ | $0.1369 \pm 0.0209$ | $0.0930 \pm 0.0034$ |
| AUTOENCODER | $\mathbf{0.1433 \pm 0.0083}$ | $\mathbf{0.0639 \pm 0.0032}$ | $\mathbf{0.1227 \pm 0.0135}$ | $\mathbf{0.0671 \pm 0.0030}$ |

Table 1: MAE comparison between synthetic and generated calcium signals. Results of (A) identity, ResNet and AGResNet generators trained with LSGAN objective, (B) AGResNet generators trained with different objectives and (C) neurons ordered by original annotation, firing rate, pairwise correlation and autoencoder reconstruction loss. We also trained a 1D variant of AGResNet as a baseline which disregards the neuron spatial structure. Lowest values marked in bold.

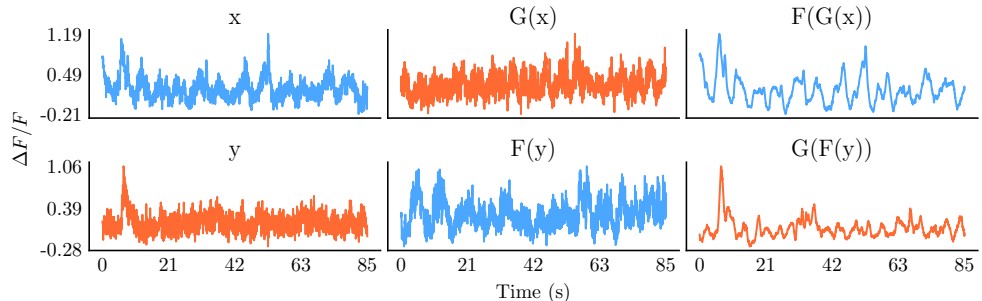

Figure 4: Forward and backward cycle steps of neuron 50 from a randomly selected test segment. More examples are shown in Figure G.1 and Figure G.2

shows that per-neuron and per-segment comparison. We first compare the firing rate distribution of each neuron between recorded and translated data (e.g. $X$ vs $F(Y)$ and $X$ vs $F(G(X))$). Examples of the distribution comparisons are available in Section H. Since we expect that the distribution of the generated data resemble of those from the recorded data, we can compare the KL divergence for each neuron to quantify the transformation performance. The firing rate distributions of $F(Y)$ and $G(X)$ closely matched the distributions of $X$ and $Y$, with average KL divergence of 1.1648 and 1.0697, respectively. Similarly, we can compute the pairwise correlation of each neuron w.r.t the population and compare the distribution between translated and recorded data. $X \mid F(Y)$ and $Y \mid G(X)$ achieved an average KL divergence value of 0.0479 and 0.0493 in the pairwise correlation comparison, both were significantly better results than the baseline identity model. In addition, we measure the van Rossum distance between $X$ and $F(Y)$ for each neuron across 200 test samples, and represent the results in the form of a heatmap. We can observe a clear diagonal line of low-intensity values in the heatmaps for most neurons (e.g. Figure H.3 and H.4 for $G$ and $F$). Hence, there exists a spike train in $X$ and $Y$ that corresponds to a translated spike train in $F(Y)$ and $G(X)$. Table G.2 summaries the average KL divergence of the 3 spike statistics in different distribution combinations, the results indicate that the generators can indeed learn the distribution translation from pre- to post-learning neuronal activities, and vice-versa. We additionally trained separate models on the activities recorded from the other mice and obtained similar results, which are available in Section I, J and K.

In the previous section, we were able to identify and interpret the learned features in a relatively straightforward manner due to the systematic augmentation we introduced into the data. However, visualizing and interpreting the attention maps on pre- and post-learning data could be more challenging as there would not be obvious patterns in the inputs to anticipate. Nevertheless, we would expect a higher level of activities in the V1 neurons when the mouse is about to enter or inside the reward zone, where the grating pattern on the virtual walls turn to black. Subsequently, the generators and discriminators should learn meaningful features from responses surrounding the reward zones. We first visualize the sigmoid masks in `AGResNet`. Figure 5 shows the learned attention masks of $G$ superimposed on the latent inputs (see Figure G.3 for $F$). When the neurons were ordered, either by firing rate or autoencoder, we observe that the generators allocate more attention toward neurons that rank higher. This suggest that by grouping neurons in a meaningful manner, the convolutional layers in the generators can extract relevant features more effectively as compared to when neurons were randomly ordered. The spike analysis showed that ordering neurons in a structured manner does indeed yield better results across the board. In most cases, ordering the neurons based on the reconstruction error achieved the best results.

We then inspect the GradCAM localization maps of the discriminators. Similar to $D_X$ in the synthetic dataset, we observed regions of high attention surrounding the reward zones in both $D_X$ and $D_Y$ (see Figure G.3 and 5). To better visualize the relationship between the area of focus learned by the model and the virtual-corridor, we generate positional attention maps as shown in Figure 6. We first compute GradCAM maps for all test samples, then we average the activation value for each neuron at each virtual position (160 cm in total) and plot the average activation value against distance. Effectively, these maps should represent the average attention learned by the models w.r.t. the visual location of the animal. Importantly, the only objective the discriminators had was to distinguish if a given sample is from a particular distribution. Thus, the discriminators could have learned trivial features. Instead, $D_X$ focused on a specific group of neurons at 100 - 130 cm in the virtual environment, which coincides with the beginning of the reward zone. Moreover, $D_Y$ learned to focus on two groups of neurons with attention patterns that were also in alignment with the reward zone in the virtual-corridor experiment. Similarly, we can extract these positional attention maps for $G(X)$ and $F(Y)$ following the same procedure, where we monitor the change in gradient in the last residual block `RB`$_9$ (bottom row in Figure 6). Interestingly, both generators focused on the first few neurons in their transformation operations. $G$ focused on activities at the beginning of the trial as well as activities in the reward zone; whereas with $F$, it paid higher level of attention to activities right before the reward zone. This suggests that to learn the transformation from post- to pre-learning responses, the activities the mouse exhibit as it approaches the reward zone is deemed more important by the networks. Note that no trial information was incorporated into the training data nor was it formulated in the objective function. Hence, these interesting patterns we observe here were learned entirely by the networks themselves via the adversarial process.

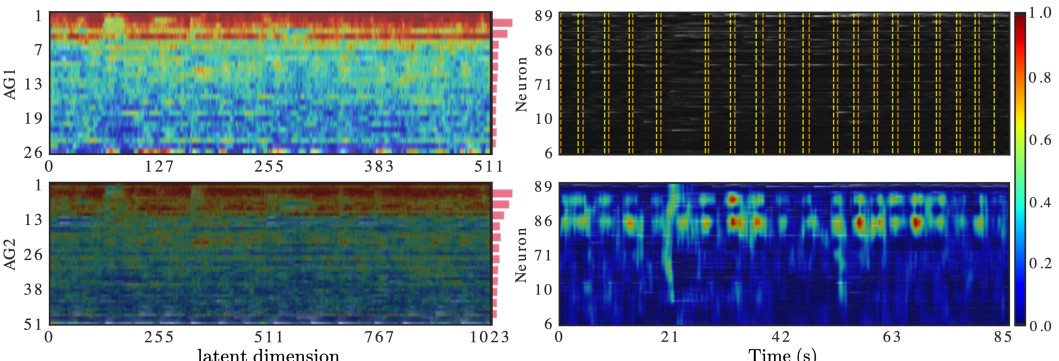

Figure 5: (Left) AG sigmoid masks in `AGResNet` $G$ given a random test segment $x \sim X$. The histograms on the right show the neuron-level attention. Note that $AG_1$ and $AG_2$ are at 4 and 2 times lower dimension than the input. (Right) GradCAM localization map of $D_Y$ given a randomly select test segment $y \sim Y$. The top panel shows the original input, where yellow and orange dotted lines mark the start and end of each reward zones. The second panel shows the localization map superimposed on the input. Neurons were ordered based on `AE` reconstruction loss. The exact ordering is available in Table E.1. Plots of $F$ and $D_X$ are shown in Figure G.3

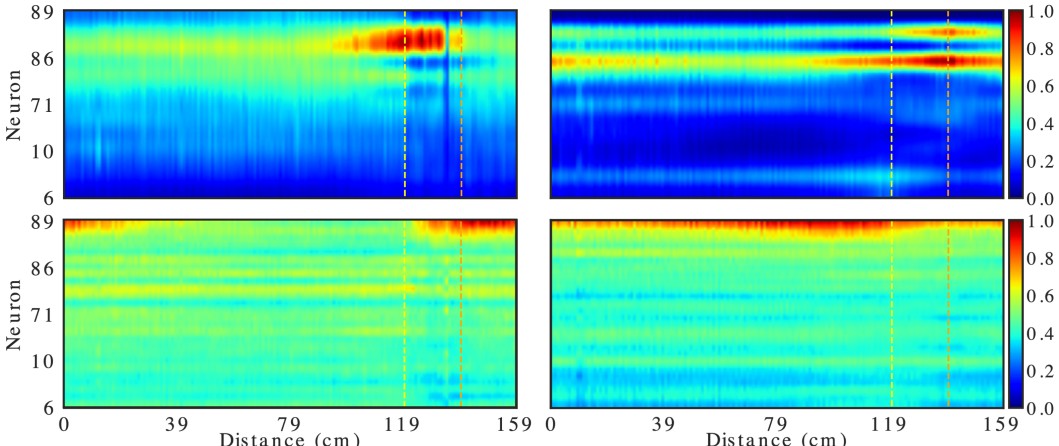

Figure 6: Positional attention maps of (Top Left) $D_X$, (Top Right) $D_Y$, (Bottom Left) $G$ and (Bottom Right) $F$ w.r.t virtual position in the animal experiment. Yellow and orange dotted lines indicate the start and end of the reward zone. Neurons were ordered based on AE reconstruction loss. The exact ordering is available in Table E.1. The pre- and post-learning activities of top 10 highlighted neurons learned by $D_X$ and $D_Y$ are available in Figure G.5 and Figure G.6b, respectively.

## 4  DISCUSSION

We demonstrated that the CycleGAN (Zhu et al., 2017) framework is a capable data-driven method to model the translation between pre- and post-learning responses recorded *in vivo*. With self-attention and feature-importance visualization methods, we are able to visualize information that the networks deemed important in their translation and discrimination process. Intriguingly, without providing trial information in the training process, the networks self-identified activities surrounding the reward zone in the virtual-corridor experiment to be highly influential, which aligns with our understanding that the responses in the visual cortex were shaped by the change of visual cues. In addition, we introduced a novel and simple to implement neuron ordering method enabling more effective learning by convolutional-based networks.

A significant portion of the neuronal activity validation in Section 3.2 was performed in spike trains inferred from the recorded and generated calcium fluorescent signals using Cascade (Rupprecht et al., 2021), which is a recently introduced method that has outperformed existing model-based algorithms. However, reliable spike inference from fluorescent calcium indicators signals remains an active area of research (Theis et al., 2016). For instance, Vanwalleghem et al. (2020) demonstrated that spiking activities could be missed due to the implicit non-negativity assumption in calcium imaging data which exists in many deconvolution algorithms, including Cascade. Nonetheless, we would like to emphasize that Cascade was used to deconvolve calcium signals for all distributions of data and therefore all inferred spike trains experienced the same bias. Another notable constraint in our method is the fundamental one-to-one mapping limitation in the CycleGAN framework. The generators learn a deterministic mapping between the two domains and only associate each input with a single output. However, most cross-domain relationships consist of one-to-many or many-to-many mappings. More recently proposed methods, such as Augmented CycleGAN (Almahairi et al., 2018), aim to address such fundamental limitations by introducing auxiliary noise to the two distributions, and are thus able to generate outputs with variations. Nevertheless, these methods are most effective when trained in a semi-supervised manner which is not possible with our unpaired neural activity.

All in all, as deep unsupervised methods have become more expressive and explainable, and neuronal activities in different learning phases from behaving animals have become more readily available, there is potential for novel insights into fundamental learning mechanisms. Future directions include sorting neurons in 2D space, as they were recorded, such that the model can take advantage of both vertical and horizontal spatial information.

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

APPENDIX

## A   ANIMAL EXPERIMENT

| DAY | NUM. TRIALS | EXPERIMENT DURATION | AVG. TRIAL DURATION | LICKS | REWARDS |
|---|---|---|---|---|---|
| 1 | 129 | 894.73s | 6.94s | 2813 | 140 |
| 2 | 177 | 898.68s | 5.08s | 2364 | 182 |
| 3 | 192 | 897.16s | 4.67s | 2217 | 198 |
| 4 | 203 | 898.45s | 4.43s | 1671 | 213 |

Table A.1: Trial information of mouse 1 in the virtual-corridor experiment across 4 days of training, which include the number of trials, average duration of each trial, total number of licks and the total reward received by the mouse. The mouse achieved "expert" level by day 4 where it had a success rate of $> 75\%$ at the task. All data were recorded at a sampling rate of 24Hz. Note that the same mouse was used in the experiment.

| MOUSE | NUM. NEURONS | DAY 1 LICKS | DAY 1 REWARDS | DAY 4 LICKS | DAY 4 REWARDS |
|---|---|---|---|---|---|
| 1 | 102 | 2813 | 140 | 1671 | 213 |
| 2 | 59 | 1038 | 75 | 1069 | 157 |
| 3 | 21 | 919 | 98 | 1065 | 302 |
| 4 | 32 | 1239 | 192 | 2493 | 230 |

Table A.2: The number of licks and rewards the 4 mice exhibit on day 1 and 4 in the virtual-corridor experiment (see Section 2.1).

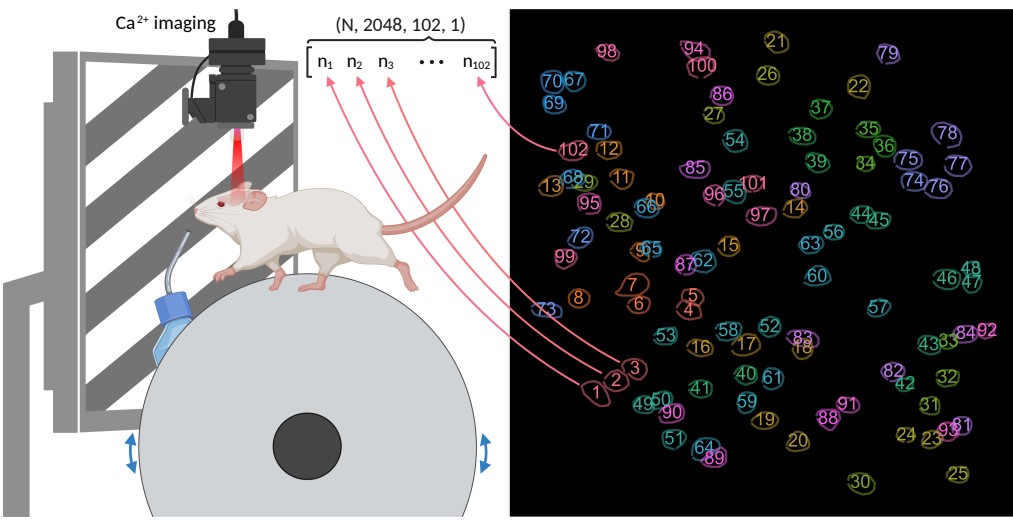

Figure A.1: (Left) illustration of the mouse virtual-environment setup. A defined grating pattern is displayed on the monitors and the mouse can move forward and backward in the virtual-corridor. When the mouse approaches the reward zone, which was set at 120 cm to 140 cm from the initial start point, the grating pattern would disappear and be replaced with a blank screen. If the mouse licked within the virtual reward zone, then a droplet of water was given to the mouse as a reward. Trials reset at 160 cm. The figure is based on Figure 1 in Pakan et al. (2018). (Right) original coordinates and annotation order of the 102 recorded neurons. i.e. neuron #1 here would be at index 0 in the data matrix, and neuron #65 would be at index 64. Neurons followed the same order across all experiments.

## B CycleGAN

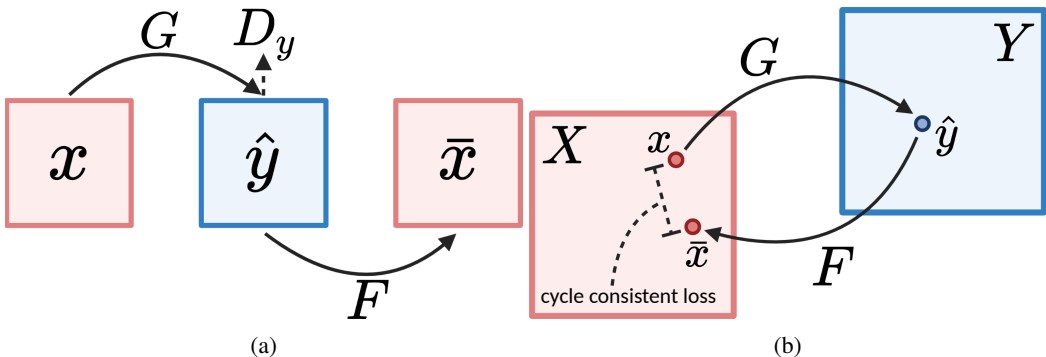

(a)                                    (b)

Figure B.1: Illustration of (a) the data flow and (b) the cycle-consistent loss in a forward cycle $X \to Y \to X$. $G$ and $F$ are generators that learn the transformation of $X \to Y$ and $Y \to X$ respectively. We first sample $x \sim X$, then apply transformation $G$ to obtain $\hat{y} = G(x)$. To ensure $\hat{y}$ resemble distribution $Y$, we train discriminator $D_Y$ to distinguish generated samples from real samples. However, even if $\hat{y}$ is of distribution $Y$, we cannot verify that $\hat{y}$ is the direct correspondent of $x$. Hence, we apply transformation $F$ which convert $\bar{x} = G(\hat{y})$ back to domain $X$. If both $F$ and $G$ are reasonable transformations, then the cycle-consistency $|x - \bar{x}|$ should be minimal. The backward cycle $Y \to X \to X$ is a mirrored but opposite operation that run concurrently with the forward cycle. Illustration re-created from Figure 3 in Zhu et al. (2017).

## C Methods

| Hyper-parameters | GAN | LSGAN | WGANGP | DRAGAN |
|---|---|---|---|---|
| Filters | | | 32 | |
| Kernel size | | | 4 | |
| Reduction factor | | | 2 | |
| Activation | | | LReLU | |
| Normalization | | | InstanceNorm | |
| Spatial Dropout | | | 0.25 | |
| Weight Initialization | | RANDOM NORMAL $\mathcal{N}(0, 0.02)$ | | |
| $\lambda_{\text{CYCLE}}$ | | | 10 | |
| $\lambda_{\text{IDENTITY}}$ | | | 5 | |
| $\lambda_{\text{GP}}$ | N/A | N/A | 10 | 10 |
| $c$ | N/A | N/A | N/A | 10 |
| NUM. DIS UPDATE | 1 | 1 | 5 | 1 |
| $\alpha_G$ | | | 0.0001 | |
| $\alpha_D$ | | | 0.0004 | |
| Distance Function | | MEAN ABSOLUTE ERROR | | |

Table C.1: The hyper-parameters used for each objective formulation. NUM. DIS UPDATE is the number of discriminator updates for every generator update, such procedure was introduced in optimizing WGANGP Arjovsky et al. (2017). $\alpha_G$ and $\alpha_D$ denotes the learning rates of the generators and discriminators. $\lambda_{\text{GP}}$ is the gradient penalty coefficient for WGANGP and DRAGAN and $c$ is the Gaussian variance hyper-parameter in DRAGAN.

| MODEL | LOSS FUNCTIONS OF $G$ AND $D_Y$ |
|---|---|

**GAN**

$$\mathcal{L}^G = - \underset{x \sim X}{\mathbb{E}} \Big[ \log(D_Y(G(x)) \Big]$$

$$\mathcal{L}^{D_Y} = - \underset{y \sim Y}{\mathbb{E}} \Big[ \log(D_Y(y)) \Big] - \underset{x \sim X}{\mathbb{E}} \Big[ \log(1 - D_Y(G(x))) \Big]$$

**LSGAN**

$$\mathcal{L}^G = - \underset{x \sim X}{\mathbb{E}} \Big[ (D_Y(G(x) - 1)^2 \Big]$$

$$\mathcal{L}^{D_Y} = - \underset{y \sim Y}{\mathbb{E}} \Big[ (D_Y(y) - 1)^2 \Big] + \underset{x \sim X}{\mathbb{E}} \Big[ D_Y(G(x))^2 \Big]$$

**WGANGP**

$$\mathcal{L}^G = - \underset{x \sim X}{\mathbb{E}} \Big[ D_Y(G(x)) \Big]$$

$$\mathcal{L}^{D_Y} = \underset{x \sim X}{\mathbb{E}} \Big[ D_Y(G(x)) \Big] - \underset{y \sim Y}{\mathbb{E}} \Big[ D_Y(y) \Big]$$

$$+ \lambda_{\text{GP}} \underset{x \sim X, y \sim Y}{\mathbb{E}} \Big[ \big( \| \nabla D(\epsilon y + (1 - \epsilon)G(x)) \|_2 - 1 \big)^2 \Big]$$

**DRAGAN**

$$\mathcal{L}^G = \underset{x \sim X}{\mathbb{E}} \Big[ \log(1 - D_Y(G(x))) \Big]$$

$$\mathcal{L}^{D_Y} = - \underset{y \sim Y}{\mathbb{E}} \Big[ \log(D_y(y)) \Big] - \underset{x \sim X}{\mathbb{E}} \Big[ \log(1 - D_Y(G(x))) \Big]$$

$$+ \lambda_{\text{GP}} \underset{y \sim Y, z \sim \mathcal{N}(0, c)}{\mathbb{E}} \Big[ \big( \| \nabla D(y + z) \|_2 - 1 \big)^2 \Big]$$

Table C.2: The objective functions of the generator $G$ and discriminator $D_Y$ in GAN (Goodfellow et al., 2014), LSGAN (Mao et al., 2017), WGANGP (Arjovsky et al., 2017) and DRAGAN (Kodali et al., 2017) formulations. The loss functions for $F$ and $D_X$ are symmetric to $G$ and $D_Y$ shown above. $\lambda_{\text{GP}}$ denotes the gradient penalty coefficient in WGANGP and DRAGAN, $\epsilon$ is the $[0, 1]$ linear interpolation coefficient for WGANGP and $c$ is the Gaussian standard deviation for DRAGAN. Note that the $\mathcal{L}^G$ listed in the table are the generator loss, and the total generator loss remains $\mathcal{L}^G_{\text{total}} = \mathcal{L}^G + \lambda_{\text{cycle}} \mathcal{L}_{\text{cycle}} + \lambda_{\text{identity}} \mathcal{L}^G_{\text{identity}}$.

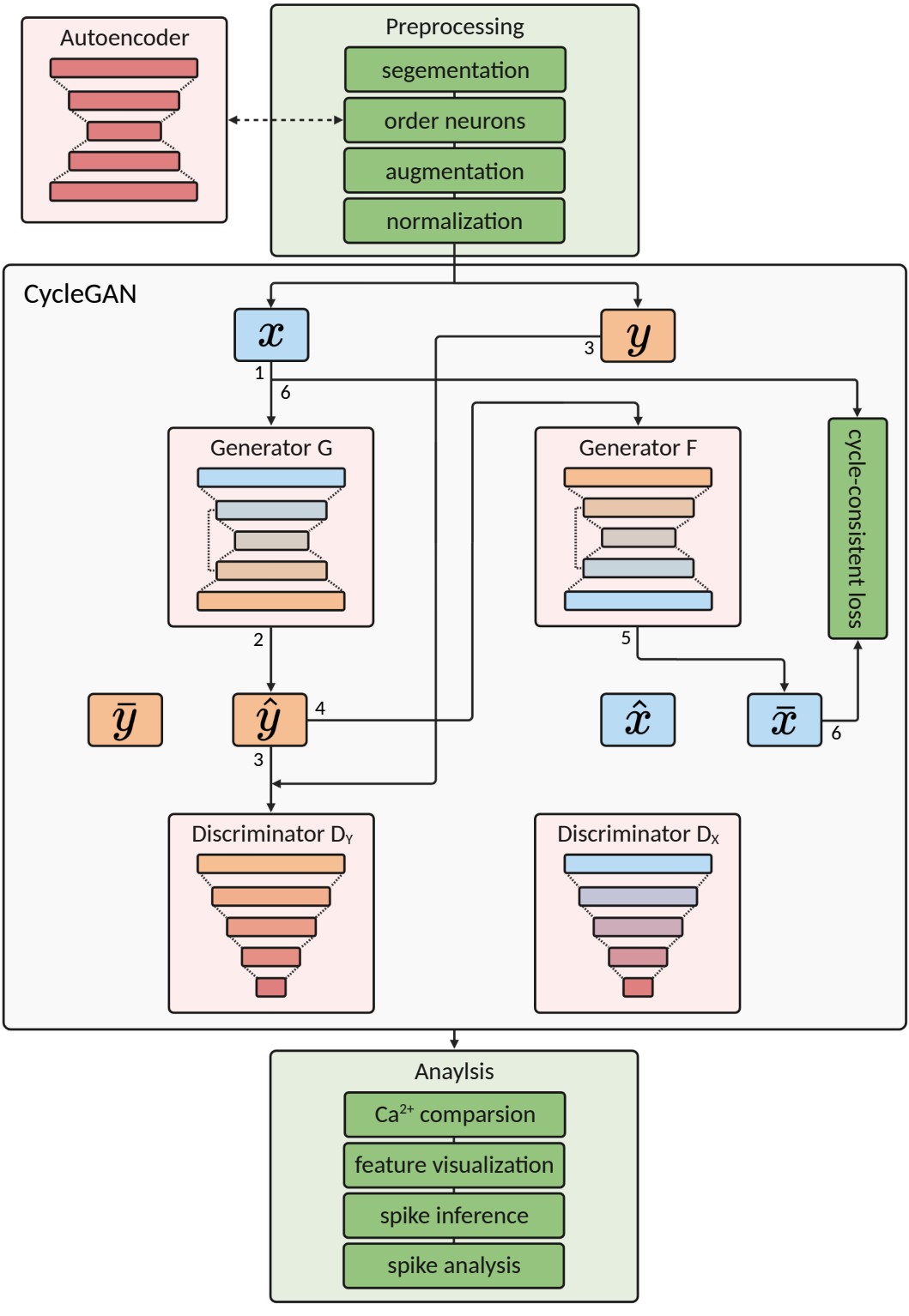

Figure C.1: Illustration of the complete pipeline used in this work. Black directed lines represent the flow of data and the numbers indicate its order. Note that only the forward cycle step $X \rightarrow Y \rightarrow X$ is shown here for better readability.

# D  NETWORKS ARCHITECTURE

The generator architecture used in this work, shown in Figure D.1, is based on the ResNet-like (He et al., 2016a) generator in CycleGAN with a number of modifications. Generally, the model consists of 2 down-sampling blocks ($DS_1$ and $DS_2$), followed by 9 residual blocks ($RB_i$ for $1 \leq i \leq 9$), then 2 up-sampling blocks ($US_1$ and $US_2$). Each down-sampling block uses a 2D strided convolution layer to reduce the spatiotemporal dimensions by factor of 2, which is then follows by Instance Normalization, LReLU activation and Spatial Dropout. Each up-sampling block has the same structure as the down-sampling blocks but with a transposed convolution layer instead. Each residual block consists of two convolution blocks with padding added to offset the dimensionality reduction and a skip connection that connect the input to the block with the output of the last convolution block via element-wise addition. A convolution layer with a filter size of 1 then compresses the channel of the output from $US_1$, followed by a sigmoid activation to scale the final output to have range $[0, 1]$.

Residual connections are known to improve gradient flow in CNN, thus mitigating the issue of vanishing gradients and allowing deeper networks to be trained effectively (He et al., 2016a;b; Huang et al., 2017). Therefore, shortcut connections are added between the down-sampling and up-sampling blocks of the same level. For instance, the output of down-sampling block $DS_2$ is concatenated with the output of residual block $RB_9$, then passes the resultant vector to the next up-sampling block $US_1$, such level-wise residual connection was first introduced in Ronneberger et al. (2015). We denote the level-wise residual connected network as `ResNet`.

Furthermore, we adapted the Additive Attention Gate (AG) module in Oktay et al. (2018) as a replacement for the concatenation operation in the residual connection described above. The yellow block in Figure D.1 illustrates the AG structure. AG takes two inputs $q$ and $a$, both with height $H_{AG}$ and width $W_{AG}$ but varying channels, where $q$ is the output of the previous processing block and $a$ is a shortcut connection from the down-sampling block of the same level. In $AG_1$ for instance, $q$ and $a$ are the output of $RB_9$ and $DS_2$ respectively. Both $q$ and $a$ are processed by two separate $1 \times 1$ convolution layers followed by Instance Normalization. The two vectors are then summed element-wise such that overlapping regions from the two vectors would have higher intensity. We then apply ReLU activation to eliminate negative values, followed by a $1 \times 1$ convolution layer with 1 filter and Instance Normalization, resulting in a vector with shape $(H_{AG}, W_{AG}, 1)$. Sigmoid activation is applied to obtain a $[0, 1]$ attention mask $\sigma$, where units closer to 1 indicate regions that are more relevant. We apply the sigmoid mask to $a$, and concatenate it with $q$. Since $q$ is a set of high-level features processed by the stack of residual blocks, whereas $a$ is the low-dimensional representation of the original input. Therefore, the sigmoid attention mask should learn to eliminate information in the input that is less relevant to the output. Moreover, as the attention mask is of the same dimension of the input $q$, we can later superimpose the attention mask onto $q$ to visualize the region of interest learned by the model. We denote the attention-gated `ResNet` as `AGResNet`.

We use a PatchGAN-based (Isola et al., 2017) discriminator architecture in this work, as it provides more fine-grained discrimination information to the generators instead of the single value discrimination in the discriminator in vanilla GAN. $D_X$ and $D_Y$ contain 3 down-sampling blocks where each block reduces the spatiotemporal dimension by a factor of 2, like the down-sampling blocks in the generators. For an input sample with shape $(H = 2048, W = 102, C = 1)$, the discriminator outputs a sigmoid activated vector with shape $(256, 13, 1)$. Each element has range $[0, 1]$ where a value closer to 1 suggests that the corresponding patch is a real sample.

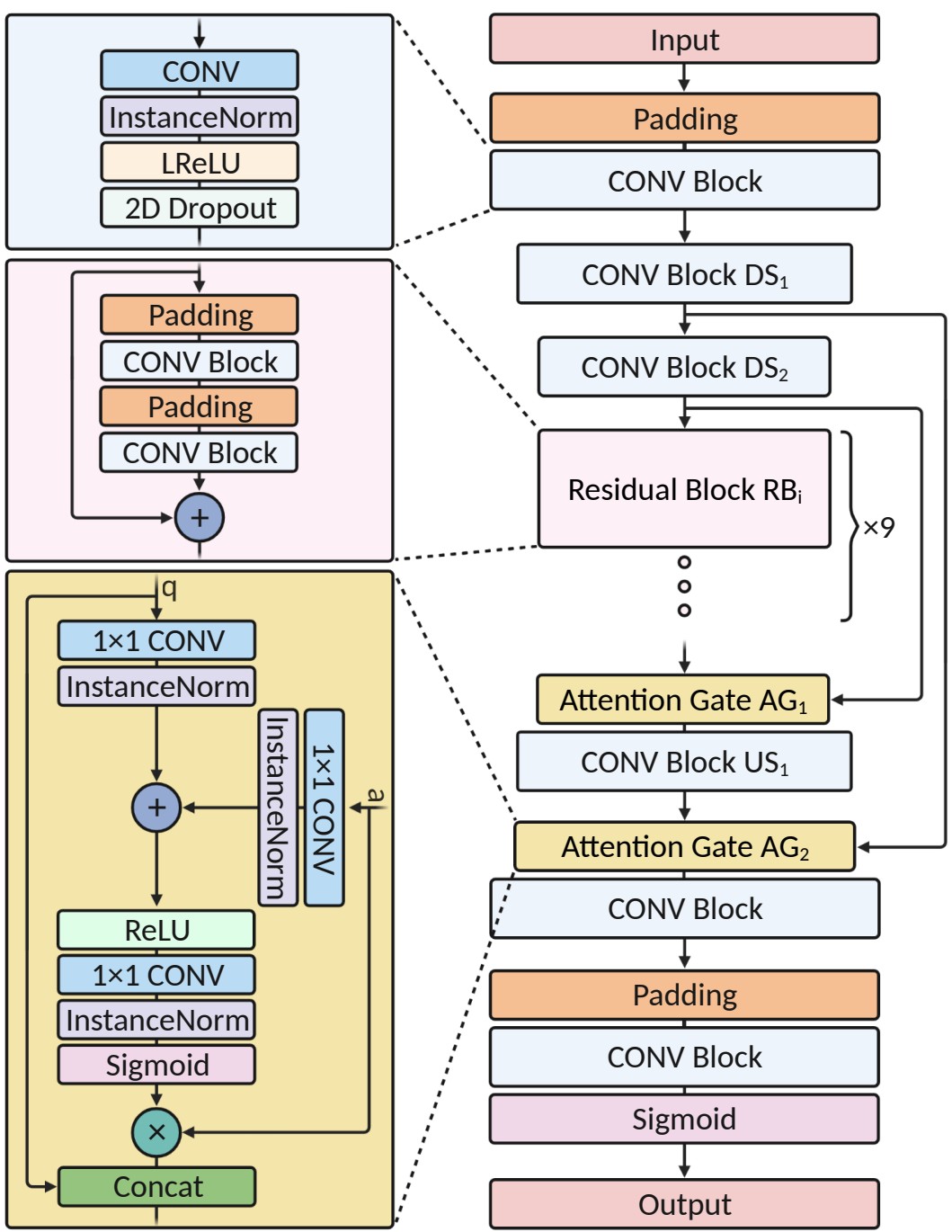

Figure D.1: Architecture diagram of generator $G$ and $F$. $+$ and $\times$ denotes addition and element-wise multiplication respectively. Note that the Attention Gate ($AG$) block can be replaced by a concatenation operation between the output of the previous block and the output from the down-sampling block from the same-level. e.g. if $AG$ is not used, then the input to $US_1$ is concat($DS_2, RB_9$).

# E  NEURON ORDERING

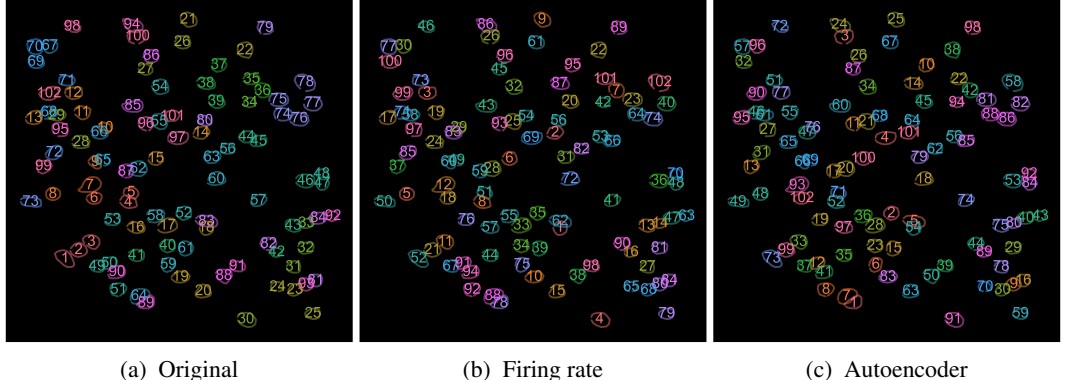

| (a) Original | (b) Firing rate | (c) Autoencoder |

Figure E.1: Neuron ordering based on (a) original annotation, (b) firing rate and (c) autoencoder reconstruction loss. The original order was based on how visible the neuron were in the calcium imaging data, hence not sorted in a particular manner. One naive approach is to sort neurons base on their overall firing rate, such that active neurons can be closer in space thus allow more efficient learning by convolutional-based networks. We proposed to train an autoencoder AE which learns to reconstruct $X$ and $Y$ jointly, and sort neurons base on the average reconstruction error on the test set. See Section 2.3.1 for detail regarding different neuron ordering methods and their motivations.

| METHOD | ORDER |
|---|---|
| (A) N/A | 1, 2, 3, 4, 5, 6, 7, 8, 9, 10, 11, 12, 13, 14, 15, 16, 17, 18, 19, 20, 21, 22, 23, 24, 25, 26, 27, 28, 29, 30, 31, 32, 33, 34, 35, 36, 37, 38, 39, 40, 41, 42, 43, 44, 45, 46, 47, 48, 49, 50, 51, 52, 53, 54, 55, 56, 57, 58, 59, 60, 61, 62, 63, 64, 65, 66, 67, 68, 69, 70, 71, 72, 73, 74, 75, 76, 77, 78, 79, 80, 81, 82, 83, 84, 85, 86, 87, 88, 89, 90, 91, 92, 93, 94, 95, 96, 97, 98, 99, 100, 101, 102 |
| (B) FIRING RATE | 18, 14, 12, 30, 8, 15, 36, 4, 21, 19, 3, 7, 43, 33, 20, 42, 13, 6, 11, 39, 2, 22, 75, 28, 55, 100, 31, 62, 10, 67, 63, 54, 17, 40, 52, 46, 99, 88, 61, 77, 57, 34, 85, 41, 27, 98, 84, 47, 65, 73, 5, 1, 44, 101, 58, 80, 16, 29, 87, 9, 26, 83, 92, 74, 24, 45, 49, 23, 97, 48, 68, 60, 71, 76, 59, 53, 70, 89, 25, 93, 32, 56, 66, 81, 72, 94, 38, 64, 79, 82, 50, 51, 96, 90, 37, 86, 95, 91, 102, 69, 35, 78 |
| (C) CORRELATION | 36, 27, 46, 28, 39, 30, 42, 20, 92, 10, 18, 11, 67, 14, 4, 33, 19, 77, 75, 13, 24, 99, 8, 43, 65, 101, 63, 7, 25, 44, 12, 76, 80, 9, 47, 3, 34, 71, 87, 52, 22, 1, 85, 61, 84, 29, 45, 31, 93, 100, 5, 58, 57, 17, 74, 21, 96, 55, 82, 91, 2, 48, 6, 56, 83, 62, 49, 16, 26, 81, 97, 53, 73, 94, 89, 59, 40, 95, 23, 32, 54, 66, 98, 72, 35, 88, 15, 41, 50, 60, 90, 70, 78, 68, 69, 86, 38, 51, 64, 79, 37, 102 |
| (D) AUTOENCODER | 89, 52, 100, 97, 83, 59, 64, 51, 93, 37, 96, 50, 99, 38, 61, 81, 87, 60, 53, 62, 55, 35, 40, 94, 21, 86, 95, 17, 32, 23, 72, 69, 3, 54, 41, 58, 49, 22, 91, 84, 90, 36, 92, 82, 39, 68, 66, 8, 73, 88, 71, 4, 46, 18, 11, 44, 70, 78, 25, 85, 29, 56, 20, 80, 28, 9, 26, 101, 65, 24, 5, 98, 1, 57, 43, 10, 12, 31, 63, 33, 75, 77, 19, 47, 45, 76, 27, 74, 42, 102, 30, 48, 7, 34, 13, 67, 16, 79, 2, 15, 14, 6 |

Table E.1: Neuron ordering based on (a) original annotation, (b) firing rate, (c) average pairwise correlation and (d) autoencoder reconstruction loss with respect to the original annotation order recorded on Day 4 data. The physical location of each neuron is available in Figure E.1.

# F    RESULTS IN SYNTHETIC DATA

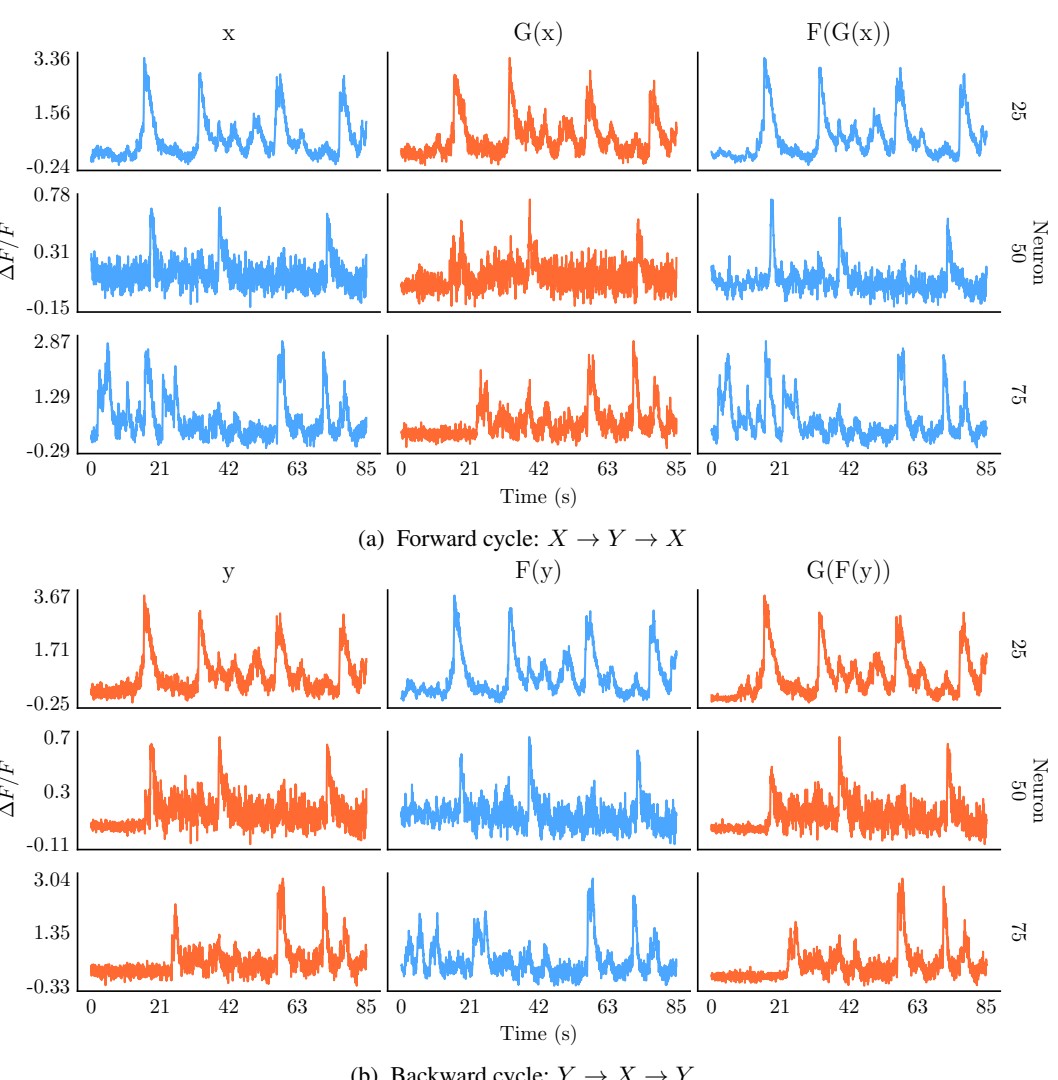

Figure F.1: (a) forward and (b) backward cycle of neuron 25, 50 and 75 from AGResNet trained with LSGAN objectives. Since $X$ and $Y$ in the test set are paired (see Section 2.3.2), we expect $x \approx F(y)$ and $y \approx G(x)$. Notice that neurons with a higher index (e.g. neuron 75 in $y$) would have more units being masked out and replaced by noise. We can see that generator $F$ was able to reconstruct the masked out regions (e.g. neuron 50 and 75 in $F(y)$) that resemble the traces in $X$.

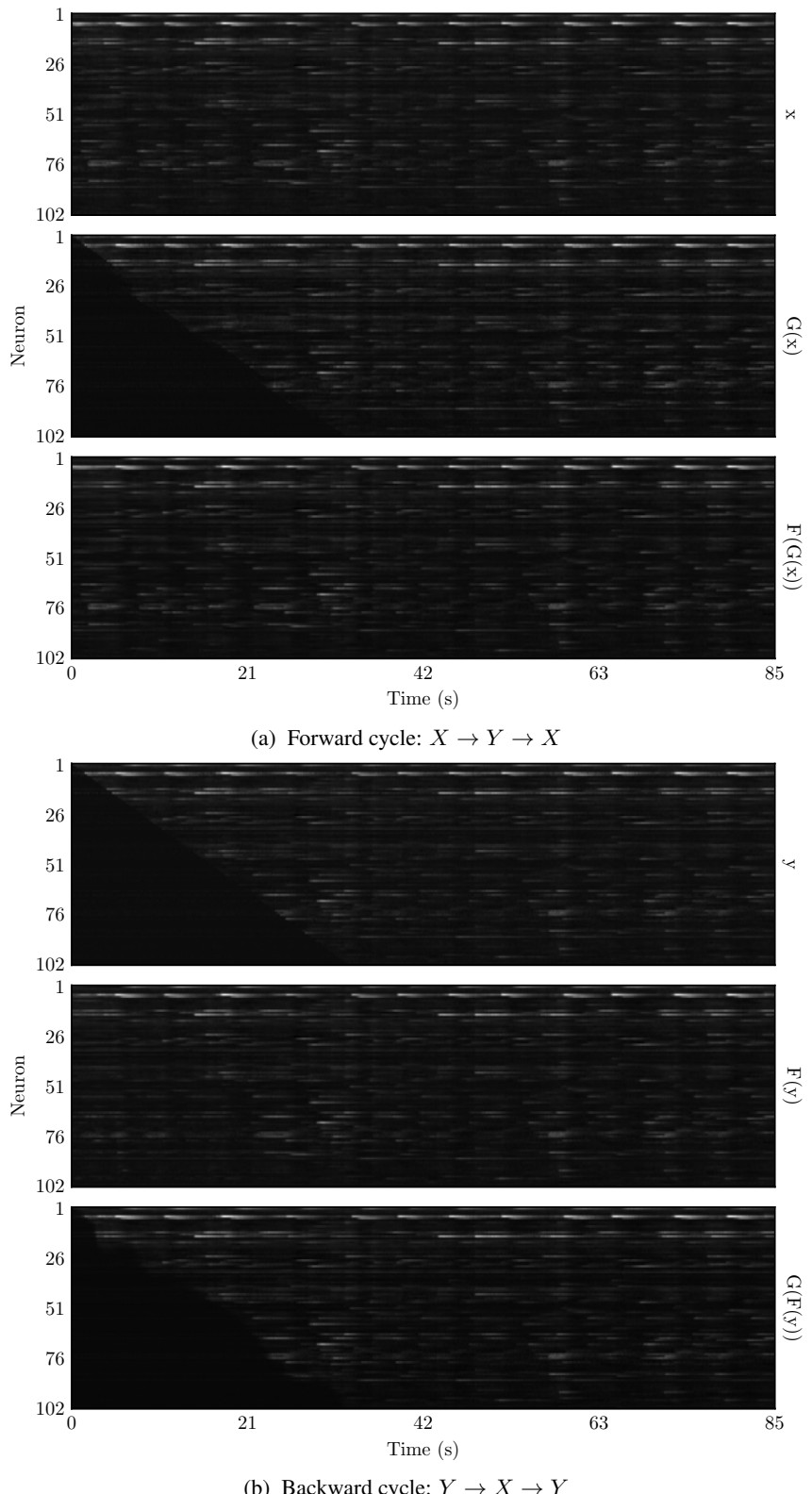

Figure F.2: (a) forward and (b) backward cycle of the entire 102 neurons from a randomly selected test segment. Model was trained with AGResNet using the LSGAN objectives on the synthetic dataset where $Y = \Phi(X)$, see Section 2.3.2 for detail.

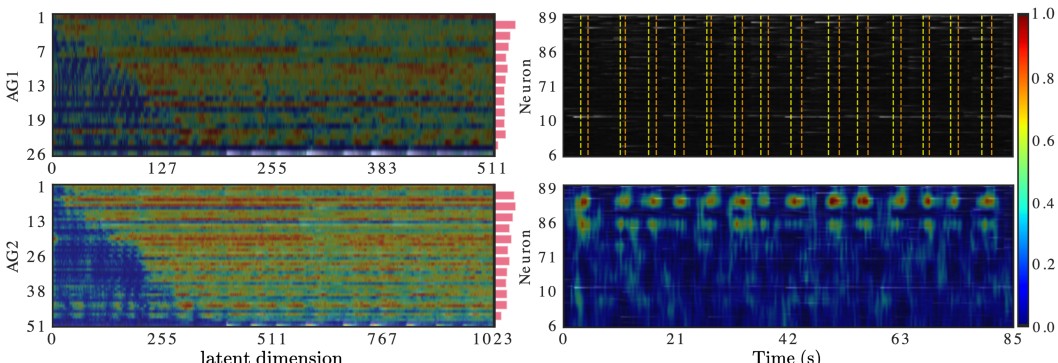

Figure F.3: (Left) Self-learned attention masks in $AG_1$ and $AG_2$ in AGResNet $G$ given a random test segment $y = \Phi(x) \sim X$. $AG_1$ and $AG_2$ were less focused in the masked region as it is filled with Gaussian noise and it not informative in the reconstruction process. Note that $AG_1$ and $AG_2$ are at 4 and 2 times lower-dimensional than the original input dimension (see Section D). (Right) GradCAM localization maps of $D_X$ given a randomly select test segment $x \sim X$. The top panel shows the original input, where yellow and orange dotted lines mark the start and end of each reward zones. The second panel shows the GradCAM localization map superimposed on the input. We observe that $D_X$ focused on neuronal activities surrounding the reward zone areas in its discrimination process, which is expected since the activities in the visual cortex are shaped by the visual-clues in the virtual-corridor experiment. Note that trial information such as reward zone locations were not provided to the networks, the pattern observed here was learned by the models themselves.

# G    RESULTS IN RECORDED DATA

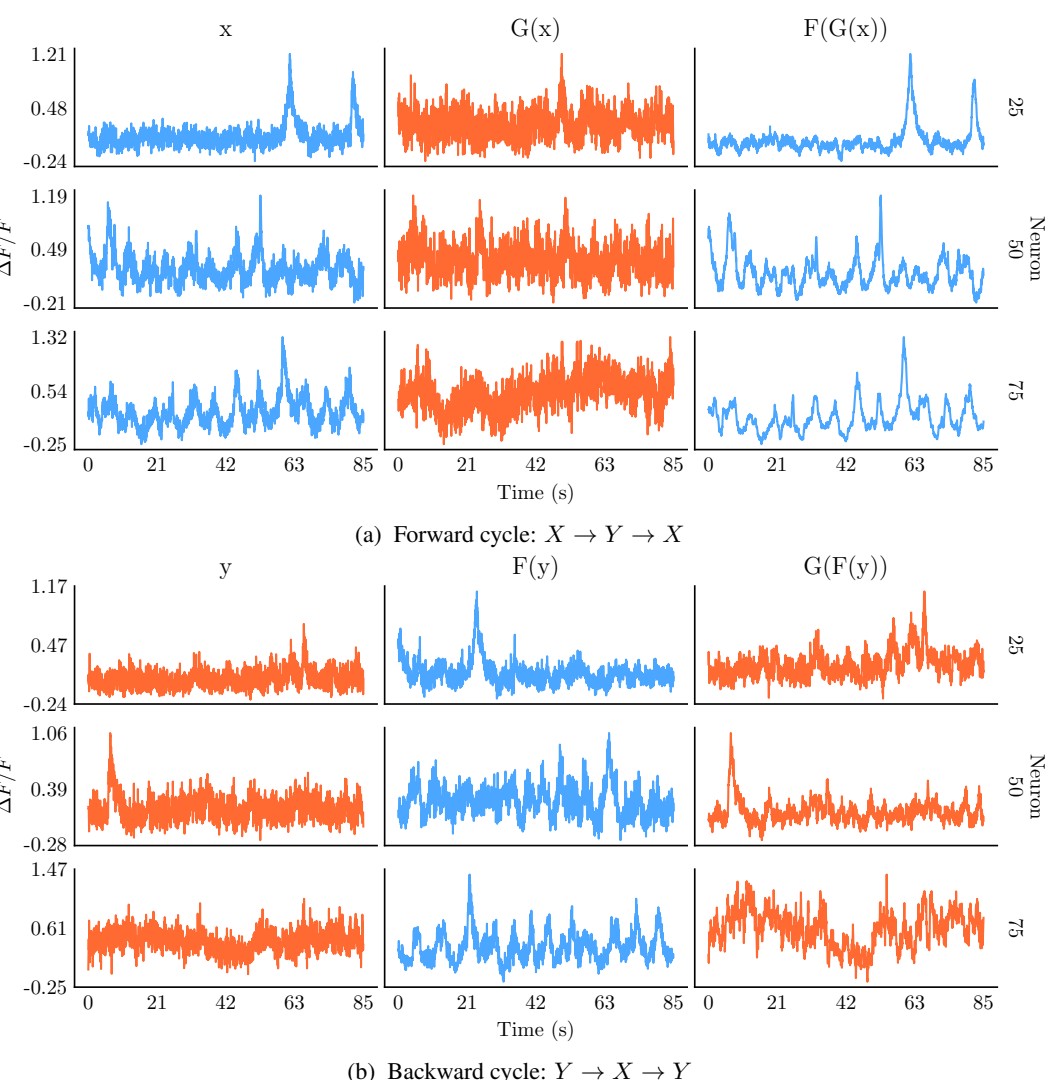

(a) Forward cycle: $X \rightarrow Y \rightarrow X$

(b) Backward cycle: $Y \rightarrow X \rightarrow Y$

Figure G.1: (a) forward and (b) backward cycle of neuron 6, 27 and 75 from a randomly selected segment. Model was trained with `AGResNet` using the LSGAN objective on the recorded dataset. Note that, unlike the synthetic dataset, the traces presented here are not unpaired. Hence, we cannot directly compare $x$ with $F(y)$ nor $y$ with $G(x)$.

| ORDER | $\lvert X - F(G(X)) \rvert$ | $\lvert X - F(X) \rvert$ | $\lvert Y - G(F(Y)) \rvert$ | $\lvert Y - G(Y) \rvert$ |
|---|---|---|---|---|
| 1D-AGResNet | $0.1806 \pm 0.0077$ | $0.1502 \pm 0.0064$ | $0.1811 \pm 0.0163$ | $0.1463 \pm 0.0149$ |
| ORIGINAL | $0.0874 \pm 0.0037$ | $0.0123 \pm 0.0015$ | $0.0766 \pm 0.0025$ | $0.0101 \pm 0.0010$ |
| FIRING RATE | $0.0760 \pm 0.0030$ | $0.0108 \pm 0.0013$ | $0.0752 \pm 0.0028$ | $0.0070 \pm 0.0005$ |
| CORRELATION | $0.0778 \pm 0.0028$ | $0.0111 \pm 0.0012$ | $0.0757 \pm 0.0024$ | $0.0089 \pm 0.0022$ |
| AUTOENCODER | $\mathbf{0.0733 \pm 0.0025}$ | $\mathbf{0.0101 \pm 0.0012}$ | $\mathbf{0.0737 \pm 0.0027}$ | $\mathbf{0.0069 \pm 0.0007}$ |

Table G.1: Cycle-consistent and identity loss in the test set of Mouse 1 recordings, where neurons were ordered by 1) original annotation, 2) firing rate 3) pairwise correlation and 4) autoencoder reconstruction loss. We also trained a 1D variant of the model as an additional baseline (`1D-AGResNet` in the table) such that all spatial information of the neurons is disregarded. The `AGResNet` generator architecture was used for $G$ and $F$, and were optimized with LSGAN objectives. The lowest loss in each category is marked in bold. For reference, $\lvert X - Y \rvert = 0.3674 \pm 0.0236$ in the test set.

| | $\mathrm{KL}(X, F(Y))$ | $\mathrm{KL}(X, F(G(X)))$ | $\mathrm{KL}(Y, G(X))$ | $\mathrm{KL}(Y, G(F(Y)))$ |
|---|---|---|---|---|
| **(A) PAIRWISE CORRELATION** | | | | |
| IDENTITY | $0.0875 \pm 0.0549$ | $\mathbf{0}$ | $0.0821 \pm 0.0471$ | $\mathbf{0}$ |
| 1D-AGResNet | $0.2027 \pm 0.1040$ | $0.4715 \pm 0.2051$ | $0.1901 \pm 0.1003$ | $0.4149 \pm 0.2194$ |
| ORIGINAL | $0.0552 \pm 0.0419$ | $0.0754 \pm 0.0353$ | $0.0583 \pm 0.0553$ | $0.0174 \pm 0.0110$ |
| FIRING RATE | $0.0507 \pm 0.0358$ | $0.0266 \pm 0.0146$ | $0.0504 \pm 0.0438$ | $0.0267 \pm 0.0176$ |
| CORRELATION | $0.0539 \pm 0.0329$ | $0.0339 \pm 0.0176$ | $0.0534 \pm 0.0474$ | $0.0205 \pm 0.0133$ |
| AUTOENCODER | $\mathbf{0.0479 \pm 0.0372}$ | $0.0329 \pm 0.0163$ | $\mathbf{0.0493 \pm 0.0448}$ | $0.0283 \pm 0.0206$ |
| **(B) FIRING RATE** | | | | |
| IDENTITY | $8.0705 \pm 6.5500$ | $\mathbf{0}$ | $7.7781 \pm 6.7338$ | $\mathbf{0}$ |
| 1D-AGResNet | $3.5688 \pm 3.8895$ | $7.9101 \pm 5.3517$ | $3.0572 \pm 3.1114$ | $8.3185 \pm 5.5950$ |
| ORIGINAL | $1.5401 \pm 1.2491$ | $2.0442 \pm 2.0936$ | $1.8527 \pm 1.3563$ | $1.4697 \pm 1.1412$ |
| FIRING RATE | $1.3402 \pm 1.0450$ | $1.2658 \pm 1.0784$ | $1.6994 \pm 1.4170$ | $1.4152 \pm 1.2221$ |
| CORRELATION | $1.4006 \pm 1.1079$ | $1.5450 \pm 1.0786$ | $1.4088 \pm 1.0828$ | $1.4674 \pm 1.3505$ |
| AUTOENCODER | $\mathbf{1.1648 \pm 0.7934}$ | $1.4022 \pm 1.2734$ | $\mathbf{1.0697 \pm 0.7689}$ | $1.2705 \pm 1.1148$ |
| **(C) PAIRWISE VAN ROSSUM DISTANCE** | | | | |
| IDENTITY | $0.5510 \pm 0.2960$ | $\mathbf{0}$ | $0.3053 \pm 0.1211$ | $\mathbf{0}$ |
| 1D-AGResNet | $0.3613 \pm 0.1597$ | $0.8045 \pm 0.1846$ | $0.3764 \pm 0.1565$ | $1.3897 \pm 0.8256$ |
| ORIGINAL | $0.2790 \pm 0.2186$ | $0.1878 \pm 0.0477$ | $0.3216 \pm 0.1352$ | $0.1581 \pm 0.0664$ |
| FIRING RATE | $0.2539 \pm 0.1708$ | $0.1003 \pm 0.0514$ | $0.3080 \pm 0.1173$ | $0.1536 \pm 0.0663$ |
| CORRELATION | $0.2629 \pm 0.1877$ | $0.1905 \pm 0.0485$ | $\mathbf{0.2953 \pm 0.1230}$ | $0.1797 \pm 0.0696$ |
| AUTOENCODER | $\mathbf{0.2387 \pm 0.1488}$ | $0.1041 \pm 0.0376$ | $0.3031 \pm 0.1138$ | $0.1328 \pm 0.0592$ |

Table G.2: The average KL divergence between generated and recorded distributions of Mouse 1 in (a) pairwise correlation, (b) firing rate and (c) pairwise van Rossum distance. We trained `AGResNet` with neurons ordered according to the following methods: 1) original annotation, 2) firing rate, 3) pairwise correlation and 4) autoencoder reconstruction loss. We also trained a 1D variant of the model (denoted as `1D-AGResNet`) such that all spatial information of the neurons is disregarded. Note that we added the identity model (first row of each sub-table) as a baseline where we should obtain perfect cycle reconstruction. Entries with the lowest value are marked in bold.

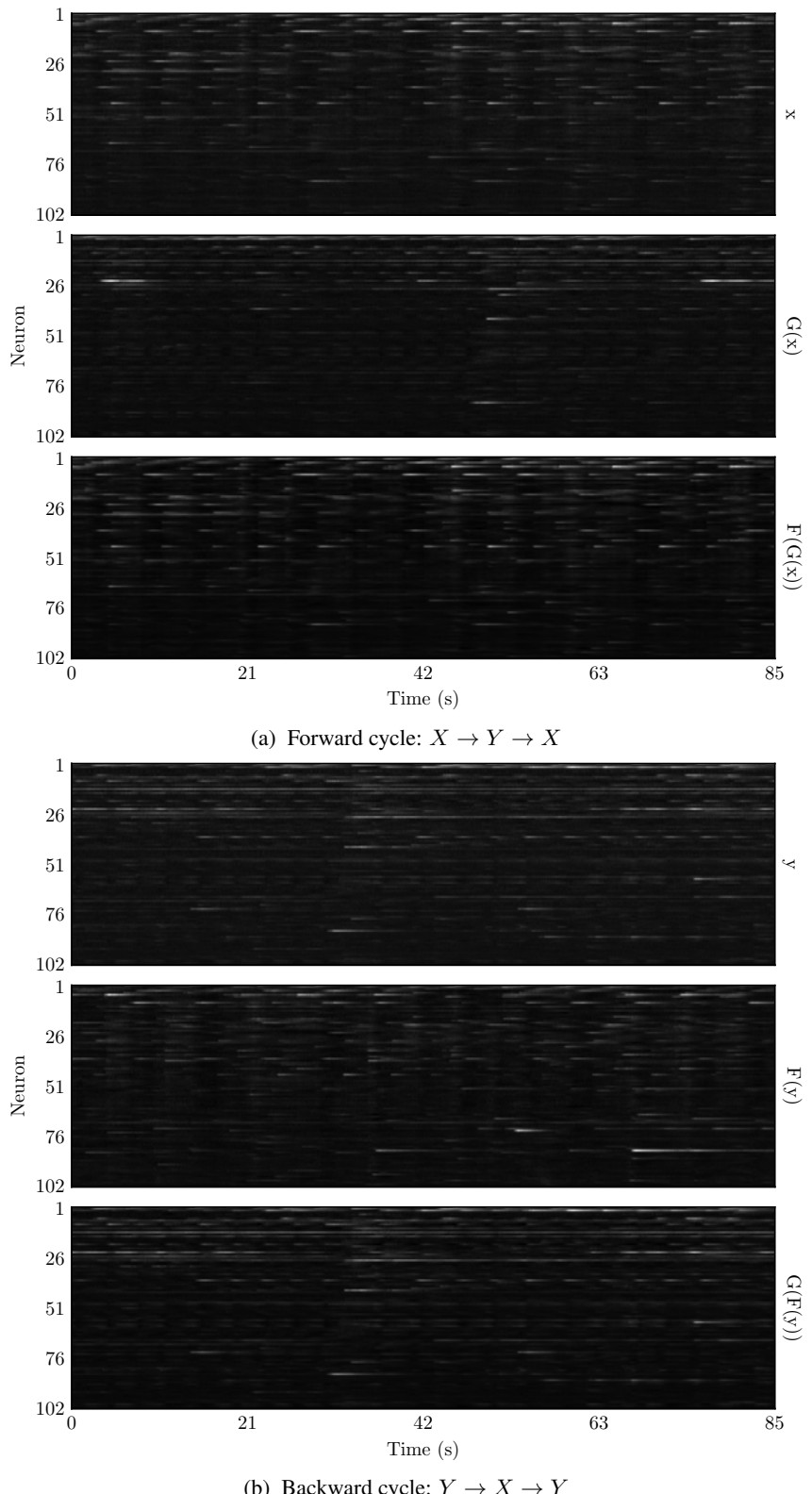

(a) Forward cycle: $X \rightarrow Y \rightarrow X$

(b) Backward cycle: $Y \rightarrow X \rightarrow Y$

Figure G.2: (a) forward and (b) backward cycle of the entire 102 neurons from a randomly selected segment. Model was trained with `AGResNet` generators using the LSGAN objective on the recorded dataset.

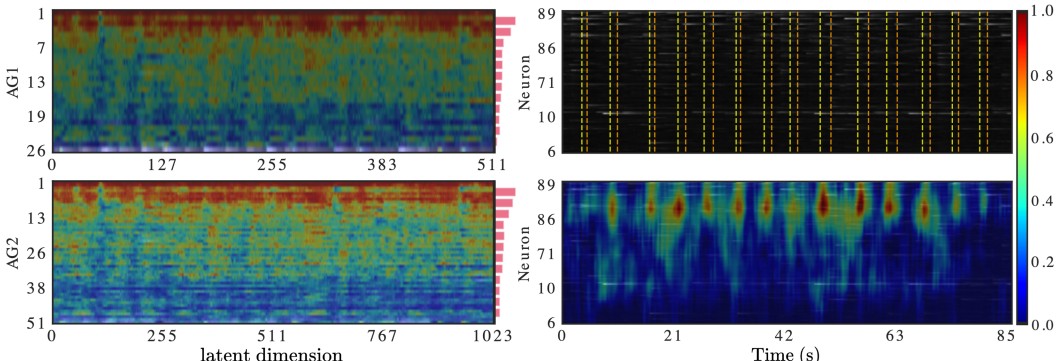

Figure G.3: (Left) self-learned sigmoid attention masks $AG_1$ and $AG_2$ in `AGResNet` $F$ given a random test segment $y \sim Y$, where neurons were sorted by the autoencoder `AE` reconstruction loss. (Right) GradCAM localization map of $D_X$ given a randomly select test segment $x \sim X$. The top panel shows the original input, where yellow and orange dotted lines mark the start and end of each reward zones. The second panel shows the localization map superimposed on the input. Again, we observe attention patterns that loosely align with the reward zones. Note that trial information such as reward zone locations were not provided to the networks, the pattern observed here was learned by the models themselves.

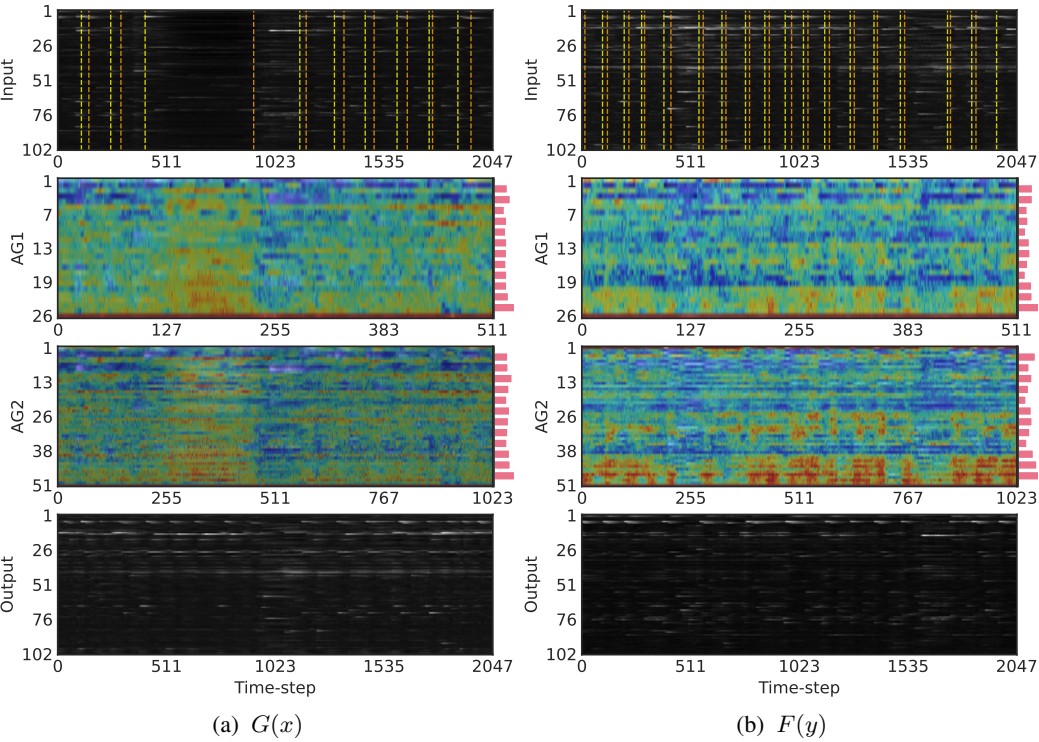

(a) $G(x)$             (b) $F(y)$

Figure G.4: Self-learned sigmoid attention masks $AG_1$ and $AG_2$ in `AGResNet` (a) $G$ given a random test segment $x \sim X$ and (b) $F$ for a given random test segment $y \sim Y$. Top panels shows the original input, where yellow and orange dotted lines mark the start and end of each reward zones. Bottom panels show the generated outputs of $G(x)$ and $F(y)$. The learned sigmoid masks shown here did not exhibit strong patterns as compare to Figure 5 and Figure G.3.

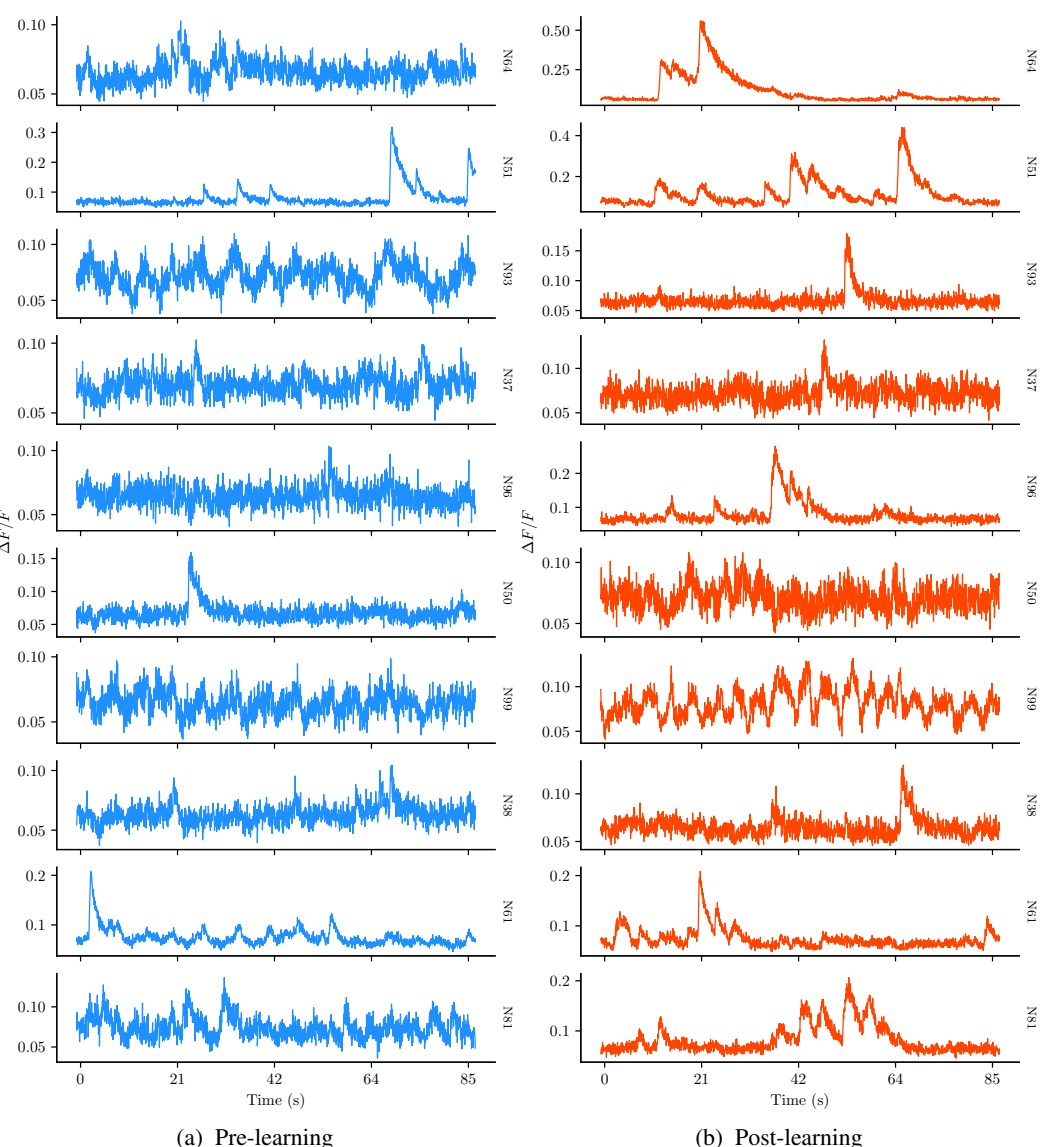

(a) Pre-learning

(b) Post-learning

Figure G.5: The (a) pre-learning and (b) post-learning traces of the top 10 neurons that $D_X$ paid the most attention to. The positional attention maps of $D_X$ is available in Figure 6 (Top Left). Note that the pre-learning and post-learning activities are not paired.

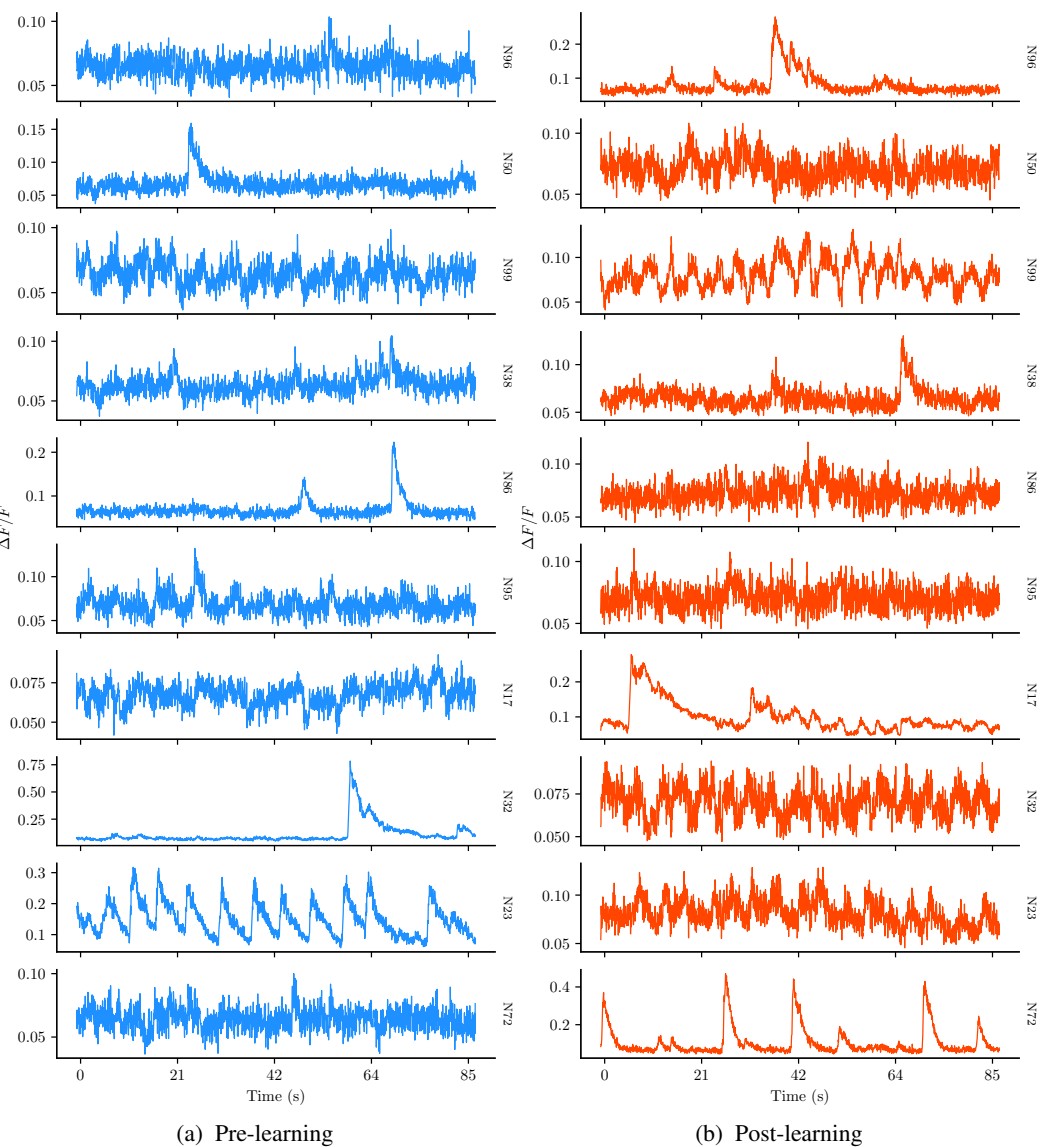

(a) Pre-learning

(b) Post-learning

Figure G.6: The (a) pre-learning and (b) post-learning traces of the top 10 neurons that $D_Y$ paid the most attention to. The positional attention maps of $D_Y$ is available in Figure 6 (Top Right). Note that the pre-learning and post-learning activities are not paired.

# H    SPIKE ANALYSIS

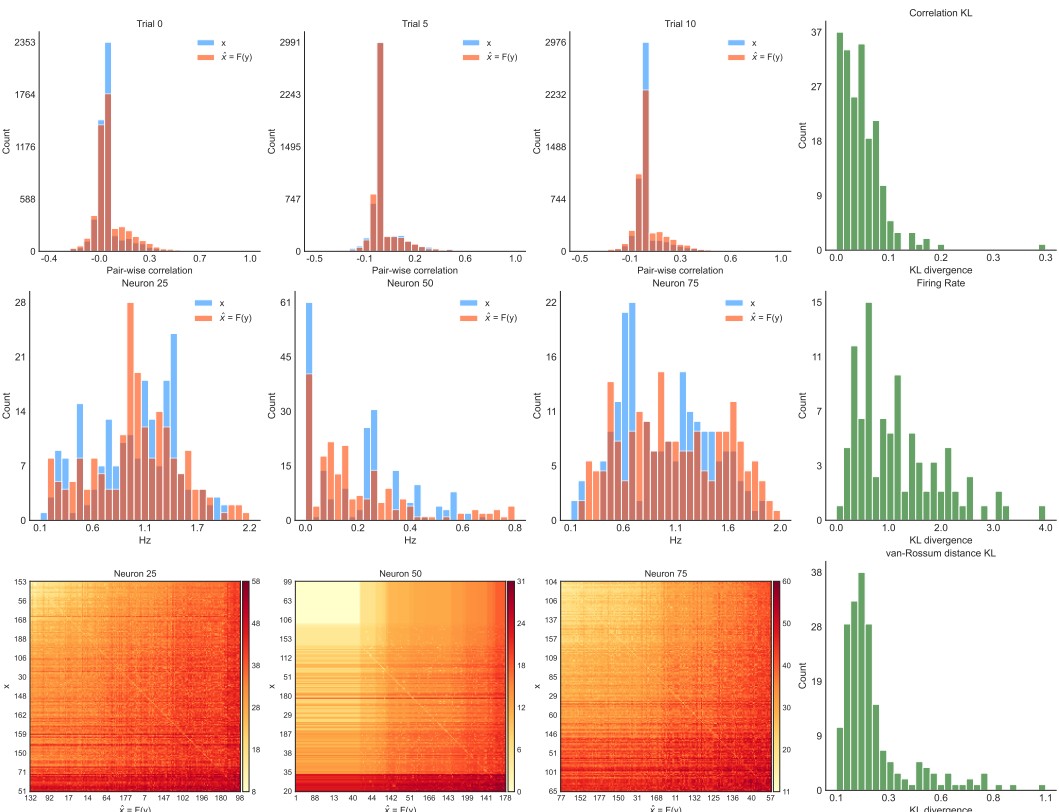

Figure H.1: Spike statistics of (top) firing rate of 3 randomly selected neurons, (middle) pairwise correlation of 3 randomly selected segments and (bottom) van Rossum distance of 3 randomly selected segments between $X$ and $F(Y)$ where $X$ was ordered by autoencoder reconstruction loss. The right columns show the KL divergence of each metrics and Table G.2 shows the mean and standard deviation of the KL divergence comparisons.

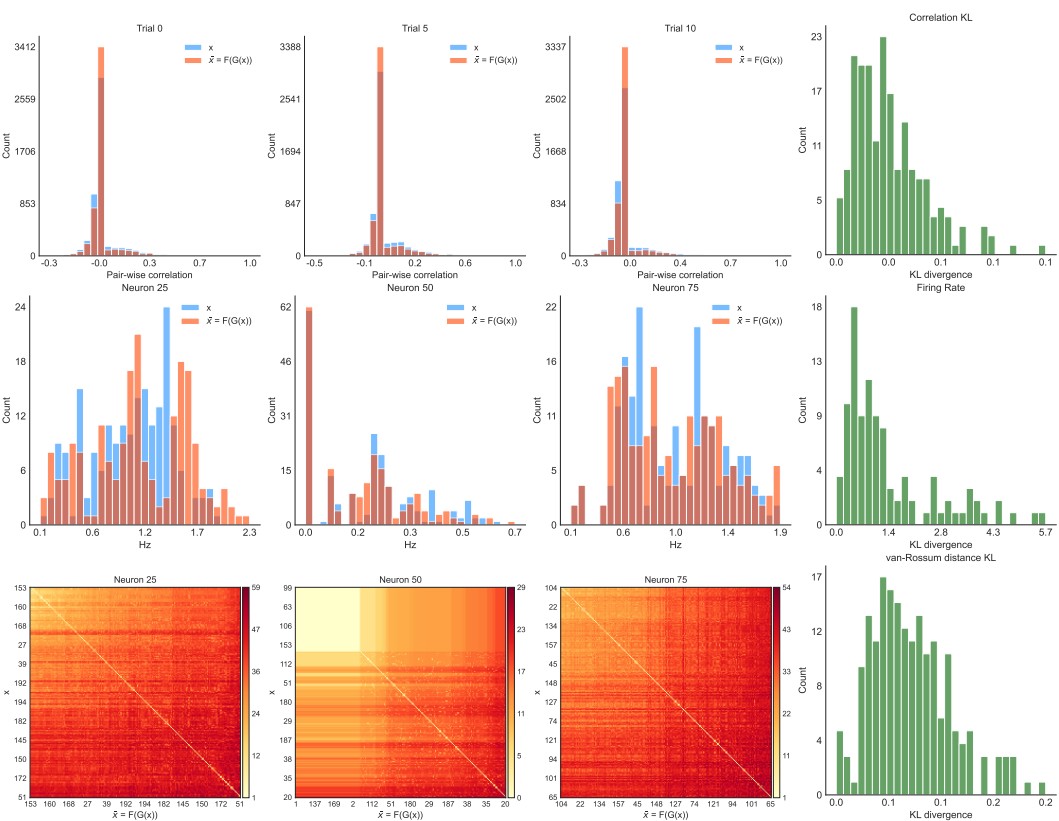

Figure H.2: Spike statistics of (top) firing rate of 3 randomly selected neurons, (middle) pairwise correlation of 3 randomly selected segments and (bottom) van Rossum distance of 3 randomly selected segments between $X$ and $F(G(X))$ where $X$ was ordered by autoencoder reconstruction loss. The right columns show the KL divergence of each metrics and Table G.2 shows the mean and standard deviation of the KL divergence comparisons.

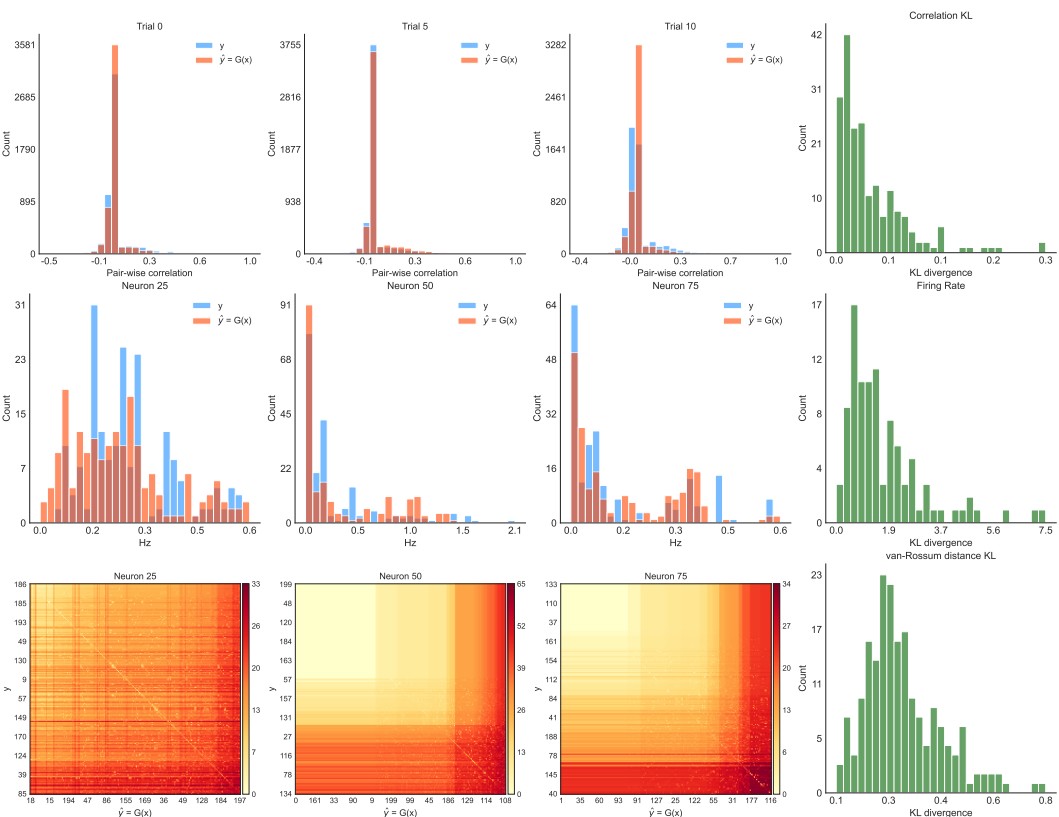

Figure H.3: Spike statistics of (top) firing rate of 3 randomly selected neurons, (middle) pairwise correlation of 3 randomly selected segments and (bottom) van Rossum distance of 3 randomly selected segments between $Y$ and $G(X)$ where $Y$ was ordered by autoencoder reconstruction loss. The right columns show the KL divergence of each metrics and Table G.2 shows the mean and standard deviation of the KL divergence comparisons.

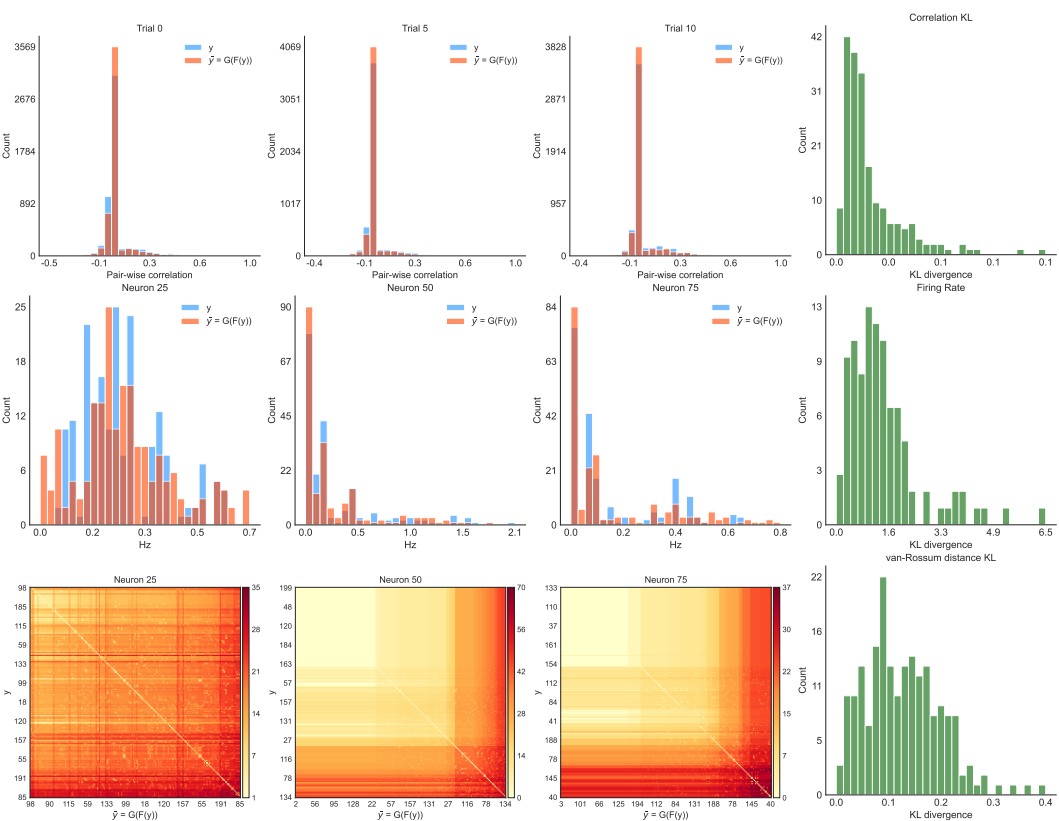

Figure H.4: Spike statistics of (top) firing rate of 3 randomly selected neurons, (middle) pairwise correlation of 3 randomly selected segments and (bottom) van Rossum distance of 3 randomly selected segments between $Y$ and $G(F(Y))$ where $Y$ was ordered by autoencoder reconstruction loss. The right columns show the KL divergence of each metrics and Table G.2 shows the mean and standard deviation of the KL divergence comparisons.

# I   Mouse 2 recorded data

| Order | $\|X - F(G(X))\|$ | $\|X - F(X)\|$ | $\|Y - G(F(Y))\|$ | $\|Y - G(Y)\|$ |
|---|---|---|---|---|
| None | $0.5875 \pm 0.1050$ | $0.1292 \pm 0.0168$ | $0.4416 \pm 0.0763$ | $0.0923 \pm 0.0064$ |
| Firing rate | $0.5794 \pm 0.1055$ | $0.1276 \pm 0.0152$ | $0.4396 \pm 0.0793$ | $0.0894 \pm 0.0048$ |
| Autoencoder | $\mathbf{0.5692 \pm 0.1008}$ | $\mathbf{0.1030 \pm 0.0099}$ | $\mathbf{0.4378 \pm 0.0769}$ | $\mathbf{0.0101 \pm 0.0018}$ |

Table I.1: Cycle-consistent and identity loss of `AGResNet` on Mouse 2 recordings, where neurons were ordered by 1) original annotation, 2) firing rate and 3) autoencoder reconstruction loss. For reference, $\mid X - Y \mid = 0.6057 \pm 0.1146$ in the test set. The lowest loss in each category marked in bold.

| | $KL(X, F(Y))$ | $KL(X, F(G(X)))$ | $KL(Y, G(X))$ | $KL(Y, G(F(Y)))$ |
|---|---|---|---|---|
| (A) **PAIRWISE CORRELATION** | | | | |
| Identity | $0.6528 \pm 0.4980$ | $\mathbf{0}$ | $0.4583 \pm 0.4366$ | $\mathbf{0}$ |
| N/A | $0.5523 \pm 0.4251$ | $0.1617 \pm 0.0715$ | $0.1212 \pm 0.0833$ | $0.0499 \pm 0.0266$ |
| Firing rate | $0.5639 \pm 0.4679$ | $0.1951 \pm 0.1031$ | $\mathbf{0.1126 \pm 0.0831}$ | $0.0399 \pm 0.0231$ |
| Autoencoder | $\mathbf{0.5209 \pm 0.5554}$ | $0.0582 \pm 0.0361$ | $0.1231 \pm 0.0988$ | $0.0352 \pm 0.0228$ |
| (B) **FIRING RATE** | | | | |
| Identity | $8.3096 \pm 6.1580$ | $\mathbf{0}$ | $5.5783 \pm 5.8451$ | $\mathbf{0}$ |
| N/A | $1.2881 \pm 1.1147$ | $2.5786 \pm 2.7222$ | $1.5782 \pm 1.2217$ | $1.6722 \pm 1.3286$ |
| Firing rate | $1.2181 \pm 0.9909$ | $2.4912 \pm 2.5037$ | $1.3656 \pm 1.1475$ | $1.1767 \pm 1.0625$ |
| Autoencoder | $\mathbf{0.8087 \pm 0.5764}$ | $1.1326 \pm 1.3149$ | $\mathbf{1.2521 \pm 0.9649}$ | $1.0592 \pm 1.0722$ |
| (C) **PAIRWISE VAN ROSSUM DISTANCE** | | | | |
| Identity | $1.3894 \pm 2.0529$ | $\mathbf{0}$ | $1.1240 \pm 1.5159$ | $\mathbf{0}$ |
| N/A | $1.3392 \pm 1.6653$ | $0.5782 \pm 0.9743$ | $0.6043 \pm 0.5250$ | $0.2497 \pm 0.2443$ |
| Firing rate | $1.2464 \pm 1.7505$ | $0.5946 \pm 0.9352$ | $0.5638 \pm 0.4181$ | $0.1977 \pm 0.1234$ |
| Autoencoder | $\mathbf{0.6946 \pm 0.5687}$ | $0.1996 \pm 0.3232$ | $\mathbf{0.5287 \pm 0.3897}$ | $0.1775 \pm 0.0959$ |

Table I.2: The average KL divergence between generated and recorded distributions of Mouse 2 in (a) pairwise correlation, (b) firing rate and (c) population pairwise van Rossum distance. We compare `AGResNet` results with different neuron ordering including 1) original annotation, 2) firing rate and 3) autoencoder reconstruction loss. Note that we added the identity model (first row of each sub-table) as a baseline where we should obtain perfect cycle reconstruction. Entries with the lowest value are marked in bold.

## J    MOUSE 3 RECORDED DATA

| ORDER | $|X - F(G(X))|$ | $|X - F(X)|$ | $|Y - G(F(Y))|$ | $|Y - G(Y)|$ |
|---|---|---|---|---|
| NONE | $0.2684 \pm 0.0290$ | $0.0656 \pm 0.0037$ | $0.3229 \pm 0.0476$ | $0.0796 \pm 0.0047$ |
| FIRING RATE | $0.2679 \pm 0.0309$ | $0.0585 \pm 0.0034$ | $\mathbf{0.3192 \pm 0.0477}$ | $0.0777 \pm 0.0043$ |
| AUTOENCODER | $\mathbf{0.2677 \pm 0.0282}$ | $\mathbf{0.0554 \pm 0.0023}$ | $0.3199 \pm 0.0487$ | $\mathbf{0.0672 \pm 0.0034}$ |

Table J.1: Cycle-consistent and identity loss of `AGResNet` on Mouse 3 recordings, where neurons were ordered by 1) original annotation, 2) firing rate and 3) autoencoder reconstruction loss. For reference, $| X - Y | = 0.4764 \pm 0.1520$ in the test set. The lowest loss in each category marked in bold.

| | $\mathrm{KL}(X, F(Y))$ | $\mathrm{KL}(X, F(G(X)))$ | $\mathrm{KL}(Y, G(X))$ | $\mathrm{KL}(Y, G(F(Y)))$ |
|---|---|---|---|---|
| | (A) **PAIRWISE CORRELATION** | | | |
| IDENTITY | $1.0188 \pm 0.5731$ | $\mathbf{0}$ | $0.7363 \pm 0.3732$ | $\mathbf{0}$ |
| N/A | $0.5361 \pm 0.2817$ | $0.5678 \pm 0.3145$ | $0.6975 \pm 0.2202$ | $0.7381 \pm 0.2977$ |
| FIRING RATE | $\mathbf{0.5021 \pm 0.2596}$ | $0.5184 \pm 0.2536$ | $0.6281 \pm 0.2830$ | $0.6616 \pm 0.2850$ |
| AUTOENCODER | $0.5140 \pm 0.2538$ | $0.4751 \pm 0.2421$ | $\mathbf{0.6137 \pm 0.2997}$ | $0.4625 \pm 0.2443$ |
| | (B) **FIRING RATE** | | | |
| IDENTITY | $12.2077 \pm 7.3556$ | $\mathbf{0}$ | $12.4075 \pm 7.3156$ | $\mathbf{0}$ |
| N/A | $1.0164 \pm 0.7129$ | $1.8203 \pm 1.9280$ | $1.2904 \pm 0.9448$ | $1.4786 \pm 1.4374$ |
| FIRING RATE | $0.9371 \pm 0.6735$ | $1.7893 \pm 2.5419$ | $\mathbf{1.0712 \pm 0.7793}$ | $1.2805 \pm 1.5136$ |
| AUTOENCODER | $\mathbf{0.8936 \pm 0.5655}$ | $1.1152 \pm 0.6797$ | $1.2114 \pm 0.7281$ | $0.6928 \pm 0.4643$ |
| | (C) **PAIRWISE VAN ROSSUM DISTANCE** | | | |
| IDENTITY | $4.2704 \pm 2.0834$ | $\mathbf{0}$ | $4.9623 \pm 1.4393$ | $\mathbf{0}$ |
| N/A | $3.0412 \pm 1.8467$ | $2.0246 \pm 1.3422$ | $4.6059 \pm 2.0664$ | $3.0293 \pm 1.5854$ |
| FIRING RATE | $2.9009 \pm 1.7587$ | $1.6458 \pm 1.2375$ | $4.1910 \pm 1.7950$ | $2.8613 \pm 1.7788$ |
| AUTOENCODER | $\mathbf{2.8383 \pm 1.5942}$ | $1.4747 \pm 1.1150$ | $\mathbf{3.9709 \pm 1.7732}$ | $1.4767 \pm 1.0195$ |

Table J.2: The average KL divergence between generated and recorded distributions of Mouse 3 in (a) pairwise correlation, (b) firing rate and (c) population pairwise van Rossum distance. We compare `AGResNet` results with different neuron ordering including 1) original annotation, 2) firing rate and 3) autoencoder reconstruction loss. Note that we added the identity model (first row of each sub-table) as a baseline comparison and should obtain perfect cycle reconstruction. Entries with the lowest value are marked in bold.

# K    MOUSE 4 RECORDED DATA

| ORDER | $|X - F(G(X))|$ | $|X - F(X)|$ | $|Y - G(F(Y))|$ | $|Y - G(Y)|$ |
|---|---|---|---|---|
| NONE | $0.2538 \pm 0.0399$ | $0.0443 \pm 0.0015$ | $0.2403 \pm 0.0395$ | $0.0808 \pm 0.0061$ |
| FIRING RATE | $0.2511 \pm 0.0389$ | $\mathbf{0.0376 \pm 0.0015}$ | $0.2388 \pm 0.0406$ | $0.0764 \pm 0.0067$ |
| AUTOENCODER | $\mathbf{0.2489 \pm 0.0381}$ | $0.0382 \pm 0.0012$ | $\mathbf{0.2367 \pm 0.0396}$ | $\mathbf{0.0764 \pm 0.0053}$ |

Table K.1: Cycle-consistent and identity loss of `AGResNet` on Mouse 4 recordings, where neurons were ordered by 1) original annotation, 2) firing rate and 3) autoencoder reconstruction loss. For reference, $|X - Y| = 0.4383 \pm 0.2354$ in the test set. The lowest loss in each category marked in bold.

| | $\mathrm{KL}(X, F(Y))$ | $\mathrm{KL}(X, F(G(X)))$ | $\mathrm{KL}(Y, G(X))$ | $\mathrm{KL}(Y, G(F(Y)))$ |
|---|---|---|---|---|
| (A) **PAIRWISE CORRELATION** | | | | |
| IDENTITY | $0.3724 \pm 0.2169$ | $\mathbf{0}$ | $0.5124 \pm 0.3238$ | $\mathbf{0}$ |
| N/A | $0.2849 \pm 0.1552$ | $0.1735 \pm 0.0918$ | $0.3536 \pm 0.2541$ | $0.5750 \pm 0.2883$ |
| FIRING RATE | $0.2482 \pm 0.1502$ | $0.1478 \pm 0.0848$ | $0.3482 \pm 0.2561$ | $0.5471 \pm 0.2577$ |
| AUTOENCODER | $\mathbf{0.2096 \pm 0.1155}$ | $0.1587 \pm 0.0867$ | $\mathbf{0.3460 \pm 0.2568}$ | $0.4795 \pm 0.2457$ |
| (B) **FIRING RATE** | | | | |
| IDENTITY | $5.8031 \pm 4.8030$ | $\mathbf{0}$ | $5.1383 \pm 5.4684$ | $\mathbf{0}$ |
| N/A | $1.3062 \pm 1.0097$ | $0.6034 \pm 0.6294$ | $1.4253 \pm 1.5599$ | $2.9196 \pm 3.1077$ |
| FIRING RATE | $1.0818 \pm 0.9274$ | $0.5480 \pm 0.5043$ | $1.2120 \pm 1.2971$ | $2.8206 \pm 2.7266$ |
| AUTOENCODER | $\mathbf{1.0564 \pm 1.1415}$ | $0.5474 \pm 0.5223$ | $\mathbf{1.1570 \pm 1.0830}$ | $2.1015 \pm 2.2399$ |
| (C) **PAIRWISE VAN ROSSUM DISTANCE** | | | | |
| IDENTITY | $2.2670 \pm 1.2707$ | $\mathbf{0}$ | $2.8134 \pm 1.5536$ | $\mathbf{0}$ |
| N/A | $1.8698 \pm 1.1525$ | $0.5625 \pm 0.4399$ | $2.4011 \pm 1.4879$ | $3.3849 \pm 1.7608$ |
| FIRING RATE | $1.5416 \pm 0.9327$ | $0.3931 \pm 0.2821$ | $\mathbf{2.1379 \pm 1.4338}$ | $3.3865 \pm 1.9320$ |
| AUTOENCODER | $\mathbf{1.3246 \pm 0.8537}$ | $0.4578 \pm 0.3639$ | $2.2134 \pm 1.3838$ | $2.6537 \pm 1.6526$ |

Table K.2: The average KL divergence between generated and recorded distributions of Mouse 4 in (a) pairwise correlation, (b) firing rate and (c) population pairwise van Rossum distance. We compare `AGResNet` results with different neuron ordering including 1) original annotation, 2) firing rate and 3) autoencoder reconstruction loss. Note that we added the identity model (first row of each sub-table) as a baseline comparison and should obtain perfect cycle reconstruction. Entries with the lowest value are marked in bold.

