# OpenReview forum: "Neuronal Learning Analysis using Cycle-Consistent Adversarial Networks"
_ICLR.cc/2022/Conference — ICLR 2022 Submitted_

### Official Review · Reviewer_aJ3M · 2021-10-23

**Correctness:** 3
**Technical Novelty And Significance:** 2
**Empirical Novelty And Significance:** 3
**Recommendation:** 5
**Confidence:** 3

**Main Review:**

The approach mapping pre-learning neural activity to post-learning activity using GANs is interesting, as is the methodology to sort neurons based on an autoencoder reconstruction loss. It was also good to see the paper systematically exploring how to choose a loss function for training the GANs -- a non-trivial issue for GAN training in general, and not always addressed in GAN papers.
However, there are some major concerns about the paper:
1. The overall motivation of _why_ one would want to map pre-learning to post-learning neural activity was not clear. Although the introduction and discussion briefly discuss interpretability for neural learning, it was not clear how this analysis contributes to that.

2. Using GradCAM maps to visualise "regions of interest" for the discriminator was interesting, but this analysis does not seem to lead to any deeper understanding about the neural activity. While the discriminators learn that the activity around the reward regions are critical to distinguishing pre- and post-learning activity (Figure 5), it is not clear why the generators don't do this. Moreover, it is possible that this effect would vanish, or other regions of interest might appear if the networks were conditioned on information about the stimulus or trials. It is also not clear how this analysis is generally applicable, or what insights can be gained if it were applied to neural data from a different task.

3. Reordering the neurons in the data using an autoencoder reconstruction loss appears to be a critical preprocessing step in the training pipeline -- however, the choice of an autoencoder over other approaches to reordering are not clearly motivated. Although this does lead to better reconstruction of the neural data from the GANs, it appears to make the subsequent step with the CycleGAN redundant: if you can accurately reconstruct neural data from the autoencoder, then why train an additional set of adversarial networks to do the same thing again?

4. It would be nice to have an estimate of the compute resources and time required to train all the networks in the pipeline (autoencoder, CycleGAN) and perform the post-hoc analysis with GradCAM, etc. In the light of doubts about motivation and benefits of the method, it would also be relevant to know how computationally expensive it is to implement.

5. There was an overall lack of clarity, and particularly in the methods and results section:
 - all equations are inline and not numbered and therefore references to terms in the equation are hard to keep in mind while trying to understand section 2.2 and 2.3;
 - extensive details about architecture and training frequently detract from understanding what steps the training pipeline consist of (perhaps these should be moved to the appendix)
- there was no explanation of how to interpret the GradCAM maps in Figures 4 and 5, or what the colours / numbers mean, and the explanation of their overlap with reward regions appeared to be handwaving. It would be nice to see the pre- and post-learning neural activity for precisely those neurons that the discriminator assigns attention to in Figure 4.
- the description of the results was confusing, and many of the plots that the conclusions here rely on are in the supplementary section, making it hard for a reader to follow any reasoning based on these plots.

Minor comments:
1. Heatmaps in Figures 2 and 4 are hard to interpret without colourbars
2. Neural activity in Figure 1 is barely visible due to the colour scheme
3. The explanation for how plots in Figure 5 is generated appears only in the caption, with no elucidation in the text.
4. There does not appear to be information of recording time per trial in the main paper


**Summary Of The Paper:**

The paper proposes to learn a mapping for neural activity in the mouse visual cortex: the mapping is from neural activity before learning to neural activity after learning, and this is achieved using CycleGAN. The paper also performs additional analysis to interpret the weights learned by the generator and discriminator networks, as well as assess the quality of the networks' reconstruction of the neural activity.

**Summary Of The Review:**

The motivation for the methods presented in the paper are not clear, the conclusions are not very convincing and consequently, it is hard to judge the contributions of the paper. A lack of clarity in the methods and results section exacerbates this.

---

> ### Author Response · Authors · 2021-11-22
> **Response to Reviewer aJ3M**
>
> We thank the reviewer for their suggested improvements and comments. We have updated our manuscript based on the suggestions made.
>
> We address each point below:
>
> **Main Review**
>
> - This concern was raised by Reviewer vrSP as well. For convenience, we here copy our response:
>   -  The transformation can form the basis for follow-up studies. It can be helpful for investigating what response characteristics change with learning as it summarizes these changes in a compact form and was obtained in a fully data-driven way. In particular, the transformation can be useful for:
>      - identifying neurons that are particularly important for describing the changes in the overall response statistics (not limited to firing rate or correlation statistics). These neurons can then be analyzed in more detail to help understand the learning process.
>      - detecting relevant response patterns. The transformation highlights population activity patterns that are important for describing the changes from pre- to post-learning.
>      - determining what virtual corridor parts or more generally what experimental details are of particular interest. In the experiment that we analyzed in this paper, the transformation pointed out the reward zone indicating that it is important for learning even though we didn't tell the algorithm anything about the experimental setup. In this experiment, it is obvious that the reward zone will be of interest for learning but in other experiments that might not generally be the case.
>   -  We added a summary of these points to the introduction.
> - As mentioned in our response to the previous concern, the visualizations should be seen as a first step with follow-up analyses. In particular, the visualisations do highlight neurons that are important for describing the changes in statistics from pre- to post-learning. As such, the method does provide additional insights into the recorded dataset. If the networks were conditioned on information about the stimulus or trials then we would indeed expect that the regions of interest would change. After all, the regions highlight areas that are relevant for transforming pre- to post-learning. Conditioning would alter the relevance of those regions. However, the fact that the visualizations pointed out the reward zone of the experimental task should be seen as proof of principle that the method can detect relevant regions with or without conditioning. Generally, the analysis can provide a compact representation of the transformation between different conditions. In this case, we explored the transformation between pre- and post-learning but one could also explore the relationship between neural activity in other conditions, such as that of the animal being at position A vs being at position B in a maze. In such experiments, an explicit representation of the changes could help identify neurons and population activity patterns related to the position. We clarified our motivation in the introduction.
> - We considerably extended the baseline orderings that we compare against. We added a baseline where we order the neurons according to their average correlation coefficients. Moreover, following a suggestion by reviewer vrSP, we added a comparison to a 1D convolution version of the model (denoted as “1D-AGResNet”) with a similar number of trainable parameters. We have updated results on synthetic data in Table 1, and results on the recorded data in Table G1 and G2. Overall, models trained on sorted neurons achieved better results compared to unordered neurons and in most cases, sorting neurons according to the autoencoder reconstruction loss performed the best. Moreover, 1D-AGResNet performed significantly worse than its 2D counterparts, suggesting that the spatial structure in the neural activities is indeed important. We cannot accurately reconstruct data from the autoencoder. The autoencoder merely helps us to find an ordering which is beneficial for convolution operations. The following “AGResNet” then exploits this improved ordering.
> - We added the computational resources and time needed to train the entire pipeline to the manuscript. We train all CycleGAN models on a single NVIDIA A100 80GB which takes on average 15 hours to complete, and the autoencoder model on average takes 1 hour to train. The self-attention and GradCAM plots were generated on the fly on the CPU hence do not increase the total computation time.
> - To help keep track of the most important equations, we numbered them and refer to them in sections 2.2 and 2.3.
> - We agree with the reviewer that architecture and training details were detracting and now moved section 2.4 Network Architecture to the appendix and only briefly describe the networks in the main text.
>
> (Please see the next comment)

---

> > ### Author Response · Authors · 2021-11-22
> > **Response to Reviewer aJ3M (continue)**
> >
> > - We clarified the explanations of figures 3, 4 and 5 (see the response to Reviewer cz5b). We also added the pre- and post-learning activities of the top 10 GradCAM-highlighted neurons to Figure G.5 and G.6 in the Appendix.
> > - We now inserted short summaries at places that previously required looking up details in the appendix and hope that the paper is now self-contained on 9 pages.
> >
> > **Minor comments**
> >
> > - We added colour-bar to figures 3 and 4, as well as figures F.3 and G.3.
> > - We now use a different colour map to improve visibility.
> > - We extended the description of figure 5 in the main text.
> > - We added a comment on the average recording time per trial to the methods section. Trial information of all 4 mice is also available in Table A.1 and A.2.
> >
> > We would like to thank the reviewer for their time and constructive comments, we hope that the updated manuscript addresses the concerns raised by the reviewer.

---

> > > ### Comment · Reviewer_aJ3M · 2021-11-23
> > > **Re: revised manuscript**
> > >
> > > I thank the authors for their detailed responses, and for the updates to the manuscript, including the figures -- they are much more readable now.
> > > I am also clearer about the motivation of the paper and the analysis in the results section, and have updated my score.
> > >
> > > However, I am still confused by the pre-processing step of sorting the neurons using an auto-encoder.
> > > If the motivation is to ensure that most correlated and informative neurons are clustered together when reordered, it is not clear to me how a per-neuron mean-squared error metric for reordering them achieves this. I am also wondering whether this improvement in performance is significant, compared to using the firing rate or pairwise correlation to sort the neurons from looking at Table G1, especially given that it requires an extra hour of compute time.

---

> > > > ### Author Response · Authors · 2021-11-23
> > > > **Response to Reviewer aj3M**
> > > >
> > > > We thank the reviewer for increasing their score and for the additional time the reviewer spent on assessing our work.
> > > >
> > > > The autoencoder objective indeed minimizes the per neuron reconstruction loss. However, the convolution layers along with the bottleneck in the architecture force the autoencoder to find a suitable low-dimensional representation that exploits shared information in neighbouring neurons. The reconstruction loss of a particular neuron order, therefore, characterizes how useful the convolutions are for representing shared information. At this time, we cannot assess significance because we have single runs of the processing pipeline. However, the autoencoder ordering did yield the best overall results in terms of cycle-consistent loss and KL divergence on all 4 mice that we evaluated.

---

### Official Review · Reviewer_vrSP · 2021-10-26

**Correctness:** 4
**Technical Novelty And Significance:** 3
**Empirical Novelty And Significance:** 2
**Recommendation:** 5
**Confidence:** 3

**Main Review:**

strengths
---------
The paper proposes a novel and creative analysis method for understanding learning-related transformations in neural activity; I am not aware of work like this in the neuroscience literature.

Inclusion of self-attention is a nice way get a better handle on these potentially complex transformations. Is there more that one can say about the masks extracted in Fig. 4?

The GradCAM localization maps are also very useful for understanding what the model learns; in particular, the "positional attention maps" in Fig. 5 are really cool. It would be neat to investigate these neurons/positions in more detail, especially if you could show this model uncovered some aspect of the data that would have been difficult to find with traditional methods.


concerns
--------
Intro: "In other words, given the neural recordings of a novice animal, can we translate the neuronal activities that correspond to the animal with expertlevel performance, and vice versa?" This is a super interesting question, and the answer this paper appears to give is "yes". However, I'm left scratching my head about what *exactly* one can learn from this translation. I'm not suggesting a new analysis (though ultimately the usefulness of this method will depend on whether or not it can uncover new insights/guide new experiments), but a more thorough discussion on how these translations can be used would make this a more compelling introduction.

2.2: I found this section hard to follow; perhaps moving Fig. B.1 (or something similar) to the main text would help with this? Even more useful (but less general) would be an explanation of the CycleGAN in the context of the neural data. For example, "Let X and Y be the neural activity before and after learning, respectively. The GAN-based framework consists of a generator G: X->Y that maps novice neural activity to expert neural activity; and a second generator F: Y->X that maps expert neural activity to novice neural activity..." This would make it easier (for me at least) to have an intuitive understanding of what the different losses correspond to.

2.3: Again, describing MAE(X, F(X)) etc would be a lot clearer in the context of the neural activity

2.3.1: Why does this ordering process work? I understand that, in order to use 2D convolutions, there must be some non-random ordering to the cells, but it's a bit bizarre to me that reconstrutcion quality from an autoencoder would be meaningful in this way. Is it possible to motivate this choice better? Another useful baseline would be to just use 1D convolutions in time and remove the spatial structure. Of course this means you can no longer use architectures out of the box, but also removes a poorly understood aspect of the preprocessing.

2.3.2: What is the motivation for this spatiotemporal transformation? While useful to show that the model can handle this, it seems fundamentally different from the types of transformations present in the neural data.

2.4: There are a lot of details here that are important to document, but distract from the main point of the paper. Perhaps move most of this to supplemental and use the extra space for a model/process diagram?

3.1: The raw MAE numbers are difficult to interpret as presented in the text. Maybe one part of table E.1 could be moved to the main text? Also, I think presenting Figure 2 first is a faster way to get an intuition for what the model is doing, and how well it is working.


minor
-----
typo, second paragraph in introduction: Prince et al exteneded the framework *to* work with...

table A2: day 4 -> day 1 rewards

2.3: where do the splits 3000, 200, 200 come from? Is this the total number of segments? how did you arrive at this number, is this related to the stride of the sliding window? seems this would result in highly correlated data samples, is that an issue?

3.2: Would be nice to see parts of Fig. F.1 in the main text; maybe show a single neuron, and present more in the supplemental

**Summary Of The Paper:**

This paper presents a new method for learning the transformation in neural population activity that takes place during task learning. The method is based on CycleGAN, but includes additional modifications related to neural data and the manner in which it is collected. The paper also presents visual interrogations of the learned model to better understand details of the learning process.

**Summary Of The Review:**

The paper addresses an interesting neuroscience problem from a unique perspective; however, I am still unclear exactly what the method is learning, and how it can be used to gain additional insights into the data. My initial assessment is to not accept this paper.

---

> ### Author Response · Authors · 2021-11-22
> **Response to Reviewer vrSP**
>
> We thank the reviewer for their comments and helpful suggestions for improvement on this work. We have updated the manuscript to address some of the issues and concerns raised in the review.
>
> We address each point below:
>
> **Concerns**
>
> - We thank the reviewer for pointing out this shortcoming in the introduction and now better motivate our approach by suggesting how the transformation can form the basis for follow-up studies. The transformation can be helpful for investigating what response characteristics change with learning as it summarizes these changes in a compact form and was obtained in a fully data-driven way. In particular, the transformation can be useful for:
>     - identifying neurons that are particularly important for describing the changes in the overall response statistics (not limited to firing rate or correlation statistics). These neurons can then be analyzed in more detail to help understand the learning process.
>     - detecting relevant response patterns. The transformation highlights population activity patterns that are important for describing the changes from pre- to post-learning.
>     - determining what virtual corridor parts or more generally what experimental details are of particular interest. In the experiment that we analyzed in this paper, the transformation pointed out the reward zone indicating that it is important for learning even though we didn't tell the algorithm anything about the experimental setup. In this experiment, it is obvious that the reward zone will be of interest for learning but in other experiments that might not generally be the case.
>   - We added a summary of these points to the introduction.
> - Following the reviewer’s suggestion, we have re-written section 2.2 CycleGAN to describe the CycleGAN framework with the neural activity transformation as the example.We also updated the description of metrics in section  2.3 in the context of neural activity. We thank the reviewer for suggesting the 1D convolution in time as another useful baseline to better understand the impact of our proposed neuron pre-ordering according to autoencoder reconstruction loss. We performed the analysis on synthetic data and on recorded data with a 1D convolution version of the model (denoted as “1D-AGResNet”) with a similar number of trainable parameters. We have updated results on synthetic data in Table 1, and results on the recorded data in Table G1 and G2. Briefly, we found that the 1D convolution works significantly worse. This suggests that the spatial structure is indeed important. Moreover, we added yet another neuron pre-ordering that uses average correlation coefficients and similarly obtained better results than no ordering though worse performance than for the autoencoder ordering.
> - A more biologically plausible transformation can indeed be used instead. However, in our case, we are also interested in knowing if the models can self-extract meaningful transformation patterns from the unpaired neural activity distributions, namely using the Attention Gate module in the generators as well as visualization via GradCAM. Therefore, this particular spatiotemporal transformation allows us to interpret and visualize the generated samples easily due to its distinct augmentation pattern.
> - We moved section 2.4 Network Architecture to the appendix and now only briefly describe the networks in the main text.
> As suggested by the reviewer, we moved the synthetic direct comparison table to the main text. We now also discuss Figure 2 and the attention plots first before detailing the direct comparison results.
>
> **Minor**
>
> - The reviewer is correct: the numbers do denote the total number of segments. To clarify this, we added a sentence to the second paragraph in section 2.3 Model Pipeline. We made the stride as large as possible so that we obtained a sufficient number of samples while keeping the correlations between samples reasonably low.
> - As suggested by the reviewer, we added the transformation of one neuron from Figure F.1 to the main text.
>
> We have also addressed the typos pointed out by the reviewer.
>
> Again, we sincerely appreciate the constructive comments made by the reviewer and please feel free to make any further suggestions so that we can improve our work.

---

> > ### Comment · Reviewer_vrSP · 2021-11-30
> > **response to authors**
> >
> > I thank the authors for addressing my concerns. It is interesting that the 1D convolution performed so poorly, and that spatial structure is indeed so important. However, like reviewer aJ3M, I am still concerned about the autoencoder preprocessing. This step seems important for the proper functioning of the algorithm, and yet is not well understood in my opinion. I think further work needs to be done in this direction before I would feel confident in accepting this paper.

---

> > > ### Author Response · Authors · 2021-12-01
> > > **response to Reviewer vrSP**
> > >
> > > We thank the reviewer for their comments and suggestions. We would like to emphasize that the neuron pre-sorting procedure was proposed to improve the performance of convolutional-based networks, though was not required for our framework to function. As shown in Tables G.1 and G.2, the models trained with pre-sorted neurons (by firing rate, pairwise correlation or autoencoder) achieved measurable, though not drastically, better results than without pre-sorting neurons. Moreover, the generators and discriminators trained on neurons without pre-sorting also obtained similar positional attention maps as in Figure 6, where the networks identified activities surrounding the reward zone. We will include the positional attention maps of the networks trained without neuron pre-sorting in the camera-ready version of the paper.

---

### Official Review · Reviewer_cz5b · 2021-10-31

**Correctness:** 3
**Technical Novelty And Significance:** 1
**Empirical Novelty And Significance:** 3
**Recommendation:** 6
**Confidence:** 5

**Main Review:**

Strength
- Extensive ablation studies, although most are not in the main text but in the appendix.
- Using an autoencoder to sort neurons without bias seem to work better than sorting them by firing rate.
- Using paired synthetic data to show the effectiveness of applying CycleGAN. And for real data, cycled reconstruction and other distribution metrics are promising, interpretations from both attention masks and Grad-CAM comply well with experiment settings.

Weakness
- Figures are not explained clearly. What are 6-89 in the right columns of attention mask figures (like 3, 4, 5)?
- The writing is not very effective: for example, the second to last paragraph in section 2.4 can be simplified to a much shorter one with the addition of an equation describing it. An equation, paired with the module figure, will also be easier to understand than this long paragraph.

Question / suggestion
- Is the model shared among all mice, or does each have its individual model? Namely, is this learned mapping more universal or more individual?
- Formulation wise: self-attention can also be applied among different neurons (just like in graph attention networks), and this might eliminate the need for pre-sorting. Essentially disentangling spatial and temporal information, modeling the spatial relationship as a graph with a learned adjacency matrix. In this way, neurons will be permutation invariant/equivariant, and the sorting is not needed, and the whole model can be learned end to end.

And lastly, it’s not a 9-page paper at all: there are too many places in the experiment section referring to appendix sections that **require** one to look into them for getting a full context. The same is true for the model architecture part. I feel I’m forced to read a 32-page paper…

**Summary Of The Paper:**

This paper uses CycleGAN to map neuronal activities of mice (as measured by Calcium traces) pre- and post-learning.
The main contributions are (1) empirical results of using CycleGAN to learn the pre- and post-learning mapping look promising. (2) using both attention mask (which is for gating residual concatenations) and Grad-CAM to help with interpretation. (3) sorting neurons with an autoencoder’s reconstruction error, essentially sorting them based on their importance.

**Summary Of The Review:**

Empirically, applying CycleGAN to reveal the mapping of pre- and pos-learning neuronal activities shows good results! Although the architecture or method novelty is not significant, and there is some unclarity of the writing, it can be a good starting point for further explorations.

---

> ### Author Response · Authors · 2021-11-22
> **Response to Reviewer cz5b**
>
> We thank the reviewer for the positive assessment of our work and the helpful suggestions for improvement. We have updated the manuscript to address some of the issues and concerns raised in the review.
>
> We address each point below:
>
> **Weakness**
> - We revised the explanations of Figures 3, 4 and 5. The numbers referred to the indices of neurons. In this case, neurons were ordered according to the autoencoder reconstruction loss. The exact order can be found in Table E.1 in the updated text.
> - We revised the writing to make better use of available space. As recommended by Reviewer vrSP, we have moved section 2.4 Network Architecture to the Appendix (Section D in the updated text) as the generator architecture, though important, is not the main point of this work. We instead briefly summarize the network architecture in section 2.3 Model Pipeline and moved the result table on the synthetic dataset (Table 1 in updated text) to the main text.
>
> **Question / suggestion**
> - We now clarified in the Results section that we train one model for each mouse: The learned mapping cannot be universal since the populations of recorded neurons differ between animals. Nevertheless, we did observe a consistent pattern in that the generators and discriminators paid higher levels of attention to activities around the reward zone. The results of the individual model trained on activities recorded from the other 3 mice are qualitatively similar and shown in Section I, J and K in the Appendix.
> - We thank the reviewer for the suggestion to eliminate the need for pre-sorting by applying self-attention among different neurons. Pre-sorting has the advantage that 1) different ordering methods can be plugged in yielding an additional element for flexibility 2) the ordering itself can be interpreted and can be meaningful. For these reasons, we kept the pre-sorting approach in the present manuscript but will explore other uses of self-attention in future work.
>
> We acknowledge that the original version of the paper had many compulsory references to materials in the appendix and we apologize for the additional review burden. We now inserted short summaries at places that previously required looking up details in the appendix and hope that the paper is now self-contained on 9 pages.
>
> We thank the reviewer for their time and suggestions, and we hope the updated manuscript clarifies the concerns and suggestions raised by the reviewer.

---

### Decision · Program_Chairs · 2022-01-20

**Decision:**

Reject

**Comment:**

This paper received 2 marginally below and 1 marginally above ratings. We discussed the paper with the reviewers and there was broad consensus that 1) the paper lacked clarity; 2) multiple modeling choices were debatable (e.g., ordering or embedding of neurons and convolution over neurons!!) and not sufficiently justified (and these choices will critically impact the conclusions drawn from the analysis); 3) we were not convinced by the relevance of the synthetic data to reflect a meaningful biological process; 4) we did not see any meaningful knowledge gained for biology from this whole analysis. My recommendation is thus to reject this paper.